# Classification and properties of non-idealized coastal wind profiles – an observational study

Christoffer Hallgren[1], Johan Arnqvist[1], Erik Nilsson[1], Stefan Ivanell[1], Metodija Shapkalijevski[2], August Thomasson[1], Heidi Pettersson[3], and Erik Sahlée[1]

[1]Department of Earth Sciences, Uppsala University, Uppsala, Sweden
[2]Swedish Meteorological and Hydrological Institute, Norrköping, Sweden
[3]Finnish Meteorological Institute, Helsinki, Finland

**Correspondence:** Christoffer Hallgren (christoffer.hallgren@geo.uu.se)

**Abstract.** Non-idealized wind profiles are frequently occurring over the Baltic Sea and are important to take into consideration for offshore wind power as they affect not only the power production, but also the loads on the structure and the behavior of the wake behind the turbine. In this observational study, we classified non-idealized profiles as wind profiles having negative shear in at least one part of the lidar wind profile between 28 and 300 m: low-level jets (with a local wind maximum in the profile), profiles with a local minimum, and negative profiles. Using observations spanning over 3 years, we show that these non-idealized profiles are common over the Baltic Sea in late spring and summer, with a peak of 40% relative occurrence in May. Negative profiles (in the 28–300 m layer) were mostly occurring during unstable conditions, in contrast to low-level jets that primarily occurred in stable stratification. There were indications that the strong shear zone of low-level jets could cause a relative suppression of the variance for large turbulent eddies compared to the peak of the velocity spectra, in the layer below the jet core. Swell conditions were found to be favourable for the occurrence of negative profiles and profiles with a local minimum, as the waves fed energy into the surface layer, resulting in an increase of the wind speed from below.

## 1 Introduction

A good description and understanding of the behavior of the wind field in the lowest 300 m of the atmosphere is becoming increasingly important as the interest in wind power is rapidly growing. To optimize the power production of a wind park, it is relevant to know not only the wind speed at hub height, but also the variation of the wind speed vertically. The vertical structure of the wind profile (i.e., wind shear and wind veer) plays a major role in determining the energy content in the air flow (e.g., Elliott and Cadogan, 1990; Wagner et al., 2011), the total load on the turbine (e.g., Dimitrov et al., 2015; Gutierrez et al., 2017) and the behavior of the wake behind the turbine (e.g., Sezer-Uzol and Uzol, 2013; Gadde and Stevens, 2021; Brugger et al., 2022). In this study, only the change of wind speed with height (the wind shear) is considered.

Recent projections from the International Energy Agency (IEA, 2019) indicate that the installed capacity of offshore wind power will have to grow at an accelerating pace in the coming decades in order to meet the IEA sustainable development scenario since it is anticipated that offshore wind power will become the dominant source of electricity generation in Europe

by 2050. Offshore, the winds are generally stronger than over land and there is less horizontal and temporal variation in the wind speed, resulting in a higher net production compared to onshore turbines of similar size.

Most offshore wind parks are located in areas relatively close to the coast as this simplifies construction and maintenance and lowers the cost for connecting to the electrical grid. The Baltic Sea (Fig. 1), which is a high latitude semi-enclosed sea, is in many ways ideal for offshore wind power as the distance to the closest coastline from anywhere in the basin is always less than 150 km. The installed capacity of offshore wind turbines in the Baltic Sea is projected to grow rapidly in the coming decades: by 2050 the area could host 93 GW of offshore wind power production, compared to 2.2 GW in 2020 (COWI, 2019;

Wind Europe, 2021). However, there are many conflicting interests regarding offshore wind power production in the Baltic Sea (e.g. environmental considerations, noise and visual disturbances as well as military and transportation interests) and therefore expansion must be handled with care.

The offshore wind profile has traditionally been described as a logarithmic or power law profile, where the wind speed rapidly increases in the surface layer (the lowest tens of meters of the atmosphere) and then only weakly increases in the rest

of the Ekman layer (typically up to 0.1–1 km height). However, coastal environments – such as the Baltic Sea – are prone to having wind profiles with partly negative gradients under certain meteorological and oceanographic conditions (e.g., Smedman et al., 1996; Barthelmie et al., 2007; Svensson et al., 2016). In contrast to idealized wind profiles, the non-idealized profiles, as defined in this study, have negative shear in at least one part of the profile between 28 and 300 m. Note, also, that wind veer can cause negative gradients in the air flow perpendicular to the rotor, but that effect is not considered in this study. Partly negative

shear in a wind speed profile can lead to a local wind maximum in the profile, in the following referred to as a low-level jet (LLJ), or a local wind minimum in the profile, what we refer to as a low-level minimum (LLmin). The height of an LLJ core often appears within the height range swept by wind turbine blades, and understanding the turbulent properties at these heights is important to analyze stress on the turbine and wake effects; as well as to assess the longevity of the turbines, the extension of the wake behind a single turbine and behind the park, and the total power output from the park.

As LLJs are frequently occurring in coastal areas, they have been studied extensively using both observations, e.g. by Smedman et al. 1993 and Tuononen et al. 2017 (the Baltic Sea), Kalverla et al. 2017 and Wagner et al. 2019 (the North Sea), and Andreas et al. 2000 (the Weddell Sea), and models, e.g. by Svensson et al. 2016 and Hallgren et al. 2020 (the Baltic Sea), Kalverla et al. 2020 (the North Sea), and Nunalee and Basu 2014 and Aird et al. 2022 (the US east coast). Using measurements from a field campaign, Smedman et al. (2004) concluded that LLJs over the Baltic Sea alter the structure of the turbulence

below the jet core and attributed this to shear sheltering (Hunt and Durbin, 1999), see Sect. 2 for further explanation. A few similar studies have been performed both offshore and onshore around the globe (e.g., Prabha et al., 2008; Duarte et al., 2012; Roy et al., 2021), but the results are inconclusive regarding to what extent LLJs alter the turbulent properties of the flow and, if so, what are the driving mechanisms that lead to turbulence production.

In this study we aim to give an overview of how often and in which meteorological and oceanographic conditions non-

idealized wind profiles occur over the Baltic Sea. The study is based on observations of the wind profile between 28 and 300 m above sea level, i.e. the heights relevant for wind power, in combination with high frequency measurements of atmospheric turbulence at 10 m height and measurements of the wave conditions. With a much longer record of observations, we re-

assess the possible effect from shear sheltering as discussed by Smedman et al. (2004). In addition, not only are the turbulent characteristics of the LLJs compared to idealized profiles analyzed, but we also differentiate the LLJ cases by introducing two new groups, negative profiles and LLmins, consisting of cases with local wind speed maxima below 28 m.

The study is structured as follows: a theoretical background on the formation of LLJs, LLmins and negative profiles over the Baltic Sea is presented in Sect. 2, together with an overview of processes altering the turbulence in the atmospheric surface layer. A description of the site and the observational data used in this study is given in Sect. 3 together with a description of the methodology applied to classify and analyze the data. In Sect. 4 the results are presented, followed by a discussion in Sect. 5. A summary and some concluding remarks are given in Sect. 6.

## 2 Theory

### 2.1 Formation of non-idealized profiles

LLJs can form both day and night in any type of terrain onshore as well as offshore, both close to the coastline and far offshore. One of the first proposed mechanisms related to the formation of the nighttime LLJ forming over mid-western USA was the inertial oscillation, theoretically and mathematically explained by Blackadar (1957), see also Van de Wiel et al. 2010 for a more realistic application of the Blackadar theory within the boundary layer. During the evening transition, when the outgoing energy from the ground surface is larger than the incoming (solar) energy, the surface layer cools from below which leads to stable stratification and a suppression of turbulence. As a consequence, the turbulent transport of momentum at a given height above the ground decreases, making the pressure gradient force unbalanced. This imbalance subsequently leads to an acceleration of the wind: a process known as frictional decoupling. As the acceleration is just above the decoupled lower part of the surface layer, a maximum in the wind profile starts to form and an LLJ is created.

Similarly, frictional decoupling can also occur when warmer air is advected over a cooler surface, typically during spring or early summer when the wind is directed from land towards a water surface and the water is still cold after the winter (e.g. Smedman et al. 1993, Smedman et al. 1997 and Debnath et al. 2021) or during winter when air is advected over an ice sheet (Vihma and Brümmer, 2002). As a result of the uneven response to daytime warming of a land surface compared to the water surface, a sea-breeze circulation can form, and this alteration of the wind profile can in turn create an LLJ (e.g. Fisher 1960, see also Aird et al. 2022). In more complex terrain, LLJs can form as a result of katabatic winds in valleys (e.g., Grisogono et al., 2007) and from channeling along mountain ridges or coastlines (e.g., Ranjha et al., 2013). Also thermally driven LLJs can appear, especially ahead of cold fronts (Kotroni and Lagouvardos 1993, see also Frost 2004).

During swell, the momentum flux can be directed from the sea surface to the atmosphere (i.e. the drag coefficient is negative), if the wind is approximately aligned with the swell direction, which is the most studied case (e.g., Grachev and Fairall, 2001; Nilsson et al., 2012; Högström et al., 2018). This can result in an increase of the wind speed in the lowest tens of metres, creating a local wind maximum in the vertical profile (e.g., Hanley and Belcher, 2008; Sullivan et al., 2008; Semedo et al., 2009; Smedman et al., 2009; Wu et al., 2017). The inflow of energy from below and the increased wind speed at very low heights can in turn result in a profile having a low-level minimum in the boundary layer (Semedo et al., 2009). Hypothesizing,

negative profiles could also occur in synoptic cases of baroclinicity and a decreasing pressure gradient force with height, creating a possibly counter-acting thermal wind and additionally also in cases of downbursts or sea-breezes, increasing the wind speed more in the lowest layer than higher up. As the negative profiles and LLmins are relatively uncommon, not much research is published on these different wind profile types. We refer to studies by Kettle (2014) and Møller et al. (2020) for a description of less common wind profiles of this type.

## 2.2 Alterations of the turbulence structure in the atmospheric boundary layer

In 1999, Hunt and Durbin developed the theoretical framework for shear sheltering using the rapid distortion technique (Townsend, 1980). The theory was aimed at explaining the turbulence structure of engineering and environmental flows where the properties of the velocity fields were separated by interfaces over which the shear changed drastically. The two layers could either resonate, enhancing the turbulence, or cause shear sheltering, where perturbations due to large eddies in the outer layer would be blocked from causing perturbations in the inner layer by a stream-wise phase shift of the vertical velocity field, diminishing vertical variance close to the border between the two layers and instead increasing horizontal variance. While Hunt and Durbin (1999) considered only neutral stratification, the analytical solution of the rapid distortion equations for the stably stratified case has since been presented by both Hanazaki and Hunt (2004, only the horizontal velocity component) and Segalini and Arnqvist (2015, also only considering the horizontal components) which facilitate a better quantification of the effect of mean shear on the turbulence. During shear sheltering, only turbulent eddies of 'appropriate size' travelling with a velocity similar to the average velocity of the flow were blocked (Hunt and Durbin, 1999; Smedman et al., 2004).

Smedman et al. (2004) were the the first to adapt this theory to the atmosphere, testing if shear sheltering was present during LLJ conditions based on the assumption that in the presence of an LLJ, the boundary layer can be broadly separated into an inner layer with strong shear and an outer layer with weak shear. In the case of an LLJ, the shear profile is qualitatively different compared to non-LLJ circumstances and, as a consequence, so is also the shear production of turbulence. Indeed, Smedman et al. (2004) found indications that for the Baltic Sea LLJ, shear sheltering was occurring. The study was based on atmospheric soundings of the wind profile up to 300 m and high frequency measurements of the turbulence at approximately 10 m height. In total 174 half-hour spectra were analyzed, out of which 118 corresponded to cases with LLJs. All measurements in the analysis were performed in stable conditions with winds directed from the open sea. Analyzing the velocity spectra and the turbulent heat transfer they concluded that – in accordance with the theory of shear sheltering – there was a significant difference between cases with and without an LLJ in the profile. The results showed that both the total energy for the low frequency (large scale) eddies and the sensible heat flux at the surface was lower when an LLJ was present. The results could not be explained by the local gradients of wind speed and temperature, indicating that shear sheltering might be occurring. However, the observed results could possibly also be due to lower production of turbulence for the low frequencies owing to the shape of the non-local gradients or because the production of turbulence was larger at the spectral peak because of, for example, shear instability.

In the following study by Prabha et al. (2008), shear sheltering during nocturnal LLJs over a forested site in Maine (USA) was examined with similar conclusions as in Smedman et al. (2004): the low frequency part of the velocity spectra were suppressed at heights below the LLJ core. However, Duarte et al. (2012) questioned the applicability of shear sheltering for

atmospheric flows. Their analysis of turbulence intensity during nocturnal LLJs in stable stratification over a flat test site covered with short grass in Oklahoma (USA) not only suggested the absence of shear sheltering, but even showed an increase in turbulence intensity in the layer below the jet. Also Karipot et al. (2008) came to the conclusion that the variances and covariances were enhanced at low frequencies under the influence of LLJs, analyzing fluxes of carbon dioxide for a forested site in Florida (USA). Thomasson (2021) investigated the vertical profile of turbulence intensity during LLJ events over the

130 Baltic Sea, and concluded that during the events the turbulence intensity decreased in the layer below the LLJ core but increased in the layer above the core, compared to the average conditions before the onset of the LLJ (consistent with the theory of shear sheltering). Roy et al. (2021) found that for a coastal site in France, the nocturnal LLJ with an associated atmospheric gravity wave enhanced the turbulent kinetic energy close to the surface.

## 3   Site, Measurements, and Methods

In order to analyze the occurrence and properties of non-idealized wind profiles for a coastal site in the Baltic Sea, a data record covering 3.5 years of measurements, from 8 December 2016 to 24 June 2020, was used. In the following subsections, a site description is given followed by detailed information about the measurements of turbulence, the wind profile and the sea state. Also, the classification system for the wind profiles, the wave age and the stability of the atmospheric surface layer is presented together with a presentation of how the turbulent properties were analyzed.

### 3.1   Östergarnsholm

Östergarnsholm is a 2 km$^2$ island located 3 km east of the larger island Gotland in the Baltic Sea, see Fig. 1. Östergarnsholm is relatively flat, the terrain reaching only 0–5 m above sea level in the southern parts of the island where the measurements were performed (57° 25' 48.4" N, 18° 59' 2.9" E). In the northern and northwestern parts of the island the terrain is higher, locally up to 10–15 m above sea level. The research station has been in operation since 1995 and is presently part of the Integrated

Carbon Observation System (ICOS) with research mainly focusing on the coastal wind profiles and the transfer processes of energy and greenhouse gases between the Baltic Sea and the atmosphere (see e.g., Smedman et al., 1997; Högström et al., 2008; Gutiérrez-Loza et al., 2019; Rutgersson et al., 2020).

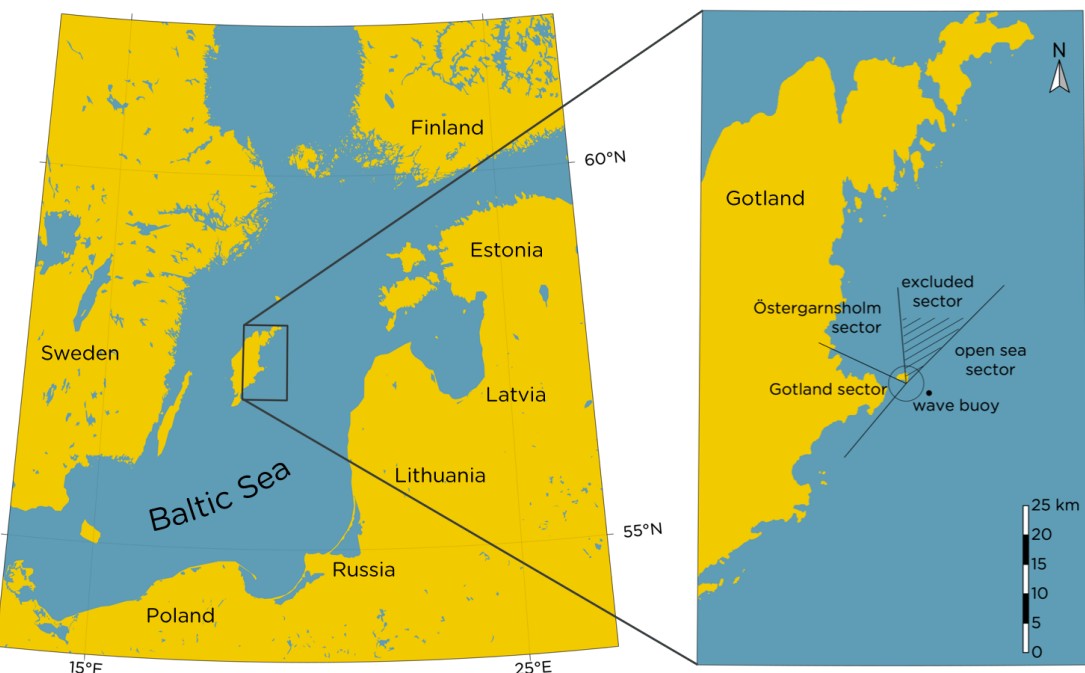

**Figure 1.** Overview of the Baltic Sea and the surrounding land areas. The circle in the inset has a radius of 3 km and is centered at the position of the meteorological mast at Östergarnsholm, the small island just east of Gotland. The open sea sector, the Gotland sector and the Östergarnsholm sector are marked in the inset together with the excluded sector from which no data was used. The position of the wave buoy is marked with the black dot in the map.

The meteorological mast at Östergarnsholm is 30 m tall and equipped with instruments measuring the temperature and wind profile. Also, turbulence measurements are performed, see further details in Sect. 3.2. Wind directions 45°–220° represent the
open sea conditions (see Fig. 1) with an undisturbed fetch of at least 150 km over the sea to the nearest coastline. In winter, sea ice can cover the northern part of the Baltic Sea, bays and coastal areas, but during none of the winters in the period December 2016 to June 2020 the maximum ice extent affected the length of the fetch in the open sea sector (SMHI, 2022). For wind directions 220°–295° the advected air comes from the Gotland sector and for 295°–355° the properties of the air are affected by Östergarnsholm. The sector 355°–45° was excluded from the analysis because of disturbances from the mast itself on the
measurements.

Approximately 30 m north of the mast, a lidar (Light Detection And Ranging) device was located, measuring the wind profile up to 300 m height, details are presented in Sect. 3.3. Located 4 km east of the mast, a wave buoy (Directional Waverider™) measured the wave field and sea surface temperature, see Sect. 3.4 for details. Only occasions when mast, lidar and wave buoy observations were simultaneously available were used in the analysis.

 **3.2 Turbulence measurements**

At the Östergarnsholm station, the main mast is an open, steel-lattice construction that has lower flow distortion properties than a mast made of a solid material. The sensors are installed on thin booms projecting 4.5 to 5 m towards the open sea sector, and the electronic units are attached as far back as possible. We restrict the analysis of turbulence measurements to wind directions between 45 and 355 degrees based on earlier studies about flow distortion and representative flux footprint areas (see Rutgersson et al. 2020 and references within).

For this study, we use high-frequency (20 Hz) wind components and temperature measured with CSAT3 three-dimensional sonic anemometers (Campbell Sci, Logan, UT, USA) at two levels, namely, 10.4 m and 26.4 m above mean sea level. We use the lowest measurement level for calculation of half-hourly mean values (wind speed, wind direction, temperature etc.) as well as second-order moments (variances and covariances), spectra and the associated stability measure, $z/L$, where $L$ is the Obukhov length (see Sect. 3.6) and $z$ is the height of the measurements. The upper measurement level, as well as additional instrumentation e.g. precipitation detection using a distrometer and received signal strength indication from the gas analyzers LI-7500 (open path) or LI-7200 (enclosed path) (LI-COR Inc., Lincoln, NE, USA), was used for initial comparison and in tests of some earlier and recently introduced quality control routines (Nilsson et al., 2018; Gutiérrez-Loza et al., 2019; Rutgersson et al., 2020) briefly discussed and summarized here.

The time series of 20 Hz data was assessed for noisy signals and non-stationarity in several ways after the sonic anemometer crosswind corrections, which are done internally. The raw high-frequency wind components were first transformed to earth-system coordinates and the angles were corrected using a double rotation method to avoid any effects caused by the tilting of the anemometer. Wind speed and wind direction were computed from the corrected wind components.

Any error flags indicating when sensors were not working properly were used to remove records prior to the calculations. A non-linear median filter algorithm was then applied to the 20 Hz data over 30 min periods to eliminate outliers from the high-frequency time series (see Brock, 1986; Starkenburg et al., 2016). By selection criterias from Vitale et al. (2020) we assure to include in our analysis only half-hours when the longest duration of gaps are less than 3 minutes and further half-hours are selected to always contain more than 85% data coverage following the SevEr thresholds suggested by Vitale et al. (2020). In practice due to additional criteria the half-hour with lowest data availability included in our analysis was 92.8% (corresponding to a total of 130 s of 20 Hz data missing in the 30 min averaging period). However, only for 0.02% of the time (ten half-hours in total), the data availability was lower than 99%. Additionally, we used the homogeneity test of fluctuations and differenced data based on Chebyshev's inequality theorem in combination with the SevEr thresholds as suggested by Vitale et al. (2020) to avoid cases of large aberrant structural changes (e.g. sudden shifts in the mean value or changes in variance) which could lead to violation of the assumption of stationarity (Vitale et al., 2020). The longest gap duration within the high-frequency time series was typically short and only 1 half-hour had more than 12 consecutive data points missing in any of our selected 20 Hz time series for wind components and sonic temperature.

Further tests were also discussed in Vitale et al. (2020), and several such criteria and choices of thresholds were initially studied. We chose here to not include the detection of poorly developed turbulence regimes, which uses the assumption that

the ratio of standard deviation of vertical wind speed and friction velocity should follow closely earlier observed measurement results in the surface layer (Mauder and Foken, 2004; Foken et al., 2012). More work is needed to reveal if this type of test is appropriate for data selection at sites that often experience low turbulence levels (e.g. low values of $\sigma_w$) and observe small friction when the flow is coming from coastal or open-sea sectors. A simpler criteria was used to remove a few cases of unrealistic low turbulence when the variance of the vertical wind speed was less than 0.0001 $\text{m}^2\,\text{s}^{-2}$.

Semi-stationary conditions were also assessed based on tests involving the non-stationarity ratio defined in Mahrt (1998) and requiring that results for second-order moments (variances and covariances) were not sensitive to being defined based on fluctuations from simple time means using block averaging or by using a linear fit over 30 min as a de-trending procedure. Non-stationarity typically increases with decreasing wind speed (Mahrt, 1998) and we chose to keep more strict limits for the stationarity tests (Vitale et al., 2020) only for higher wind speeds to be able to include sufficient data in our analysis in all wind speed intervals (see Mahrt 1998 for further discussion on this issue). At very low wind speeds we allowed a maximum on the non-stationarity ratio of 15, which could imply severe non-stationarity. However, this also allowed us to keep wind and wind stress climatology fairly intact. If we used a threshold for the non-stationarity ratio of maximum 3, as suggested in Vitale et al. (2020), an overall decrease of available 30 min statistics kept for analysis was estimated to be 16%. This may at first seem acceptable but the reduction of low wind speed data was severe: it would imply a decrease of approximately 46% for 10 m winds below 3 $\text{m}\,\text{s}^{-1}$ and a 71% decrease of data availability for wind speed conditions below 1 $\text{m}\,\text{s}^{-1}$ (winds below 3 $\text{m}\,\text{s}^{-1}$ and below 1 $\text{m}\,\text{s}^{-1}$ occurring approximately 11% and 1% of the time respectively). This would have caused severe restrictions to the analysis at the Östergarnsholm site, especially during swell with winds below 3 $\text{m}\,\text{s}^{-1}$ (these wind speeds occurring approximately 26% of the time with swell). Instead, physical reasons for non-stationarity were initially investigated and found to occur frequently during precipitation events. Signal strength quality control parameters from the gas analyzers (see Nilsson et al. 2018) as well as unusually high temperature variances was used to identify, flag and exclude suspicious outliers. Finally comparisons between sonic anemometer wind speeds, wind directions and temperature to other in-situ sensors at the site (Rutgersson et al., 2020) were used to manually flag a smaller amount of data points.

Fluxes were calculated in a rotated coordinate system (Kaimal and Finnigan 1994, natural wind coordinates with double rotation) and each 20 Hz time series that contained missing data or eliminated outliers was gap-filled with linear interpolation. The turbulent fluctuations of each variable were then calculated using a Reynold's decomposition and block averaging over 30 min periods was selected for all further analysis in this study. The turbulent fluctuations were used to calculate the variances and covariances, as well as other statistical moments used during the flux calculations and analysis. Bin-averaged spectra and cospectra for momentum and heat fluxes were also calculated for each 30 min time period (using 21 logarithmically spaced frequency bins) and analyzed together with wind lidar profile data.

### 3.3 Wind profile measurements

Vertical profiles of the wind speed were measured with a ZephIR-300 wind lidar (ZX Lidars), a conically scanning continuous wave lidar. Data from the instrument have been used before by Svensson et al. (2019) and Hallgren et al. (2020) to study the wind profile at Östergarnsholm. The unit was modified to collect raw data and had an extended range of measurements up to

300 m, similar to the current ZX300 model. The measurement cycle consisted of focusing the laser at a specific height, making three revolutions (one revolution per second) to sample the Doppler shift before moving on to the next height. In addition to the scans at each measurement height, the cycle was completed by scans without focus which, in combination with a scan at a lower height, was used for automatic data quality assessment. As such, increasing the number of measurement heights implies decreasing the amount of data available at each height for constructing the average wind profile. Reflecting the goal of detecting non-idealized wind profiles, the unit was set to measure at a relatively large number of heights; 28, 39, 50, 100, 150, 200, 250, and 300 m above sea level, which was considered a reasonable trade off between the number of heights and the statistical convergence, keeping in mind that a 30 min averaging window was used for individual wind profiles. The relatively long time averaging window reduces the relative frequency of non-idealized wind profiles compared to if a shorter time frame would have been used, see Sect. 3.5 for further comments regarding the consequences of time averaging.

The technology of optically focusing the laser beam to set the measurement height implies that the vertical extent of the measurement volume is small at low heights, but quadratically grows with height. Using a Cauchy–Lorentz distribution to determine the probability of backscatter following Mann et al. (2010; see also Svensson et al. 2019), we determined that 50% of the measurement was coming from within $\pm 0.7$ m at 28 m, $\pm 8.9$ m at 100 m and $\pm 79.9$ m at 300 m – assuming backscatter elements to be homogeneously distributed in the boundary layer and that the beam attenuation could be considered minor. However, the thick tails of the Cauchy–Lorentz distribution implied that there still is a significant probability that the measurement became contaminated with Doppler shifts from lower or higher heights when the target height increased, especially for heterogeneous distribution of aerosols, but also in ideal conditions. In practice, the wind profiles will be somewhat smoothed by this effect, particularly the non-linear variations of the wind speed in the upper part of the wind profile.

In addition to the quality control from the manufacturer, an extra quality control was performed on 30 min output averages (December 2016 – December 2017) and on 10 min output averages (January 2017 – June 2020) which were then used to calculate 30 min averages. Application of additional quality controls led to the removal of 6.7% of the data (removal of spikes, removal of profiles with data missing on two or more of the eight height levels and manual control of all profiles classified as negative, LLmin, weak LLJ or strong LLJ, see Sect. 3.5). There were two longer breaks in the lidar measurement campaign (see Fig. 2): first the removal of the lidar from the site for testing and comparison at another site (23 January – 29 April 2019), then due to service and maintenance from the manufacturer (11 August – 2 December 2019).

### 3.4  Wave measurements

Since 1995, wave measurements at 57° 25' 0.012" N, 19° 3' 11.988" E (see Fig. 1) have been performed with a Directional Waverider™ buoy, owned and run by the Finnish Meteorological Institute (FMI). The water depth at the buoy location is 39 m. The wave spectrum was calculated on board the buoy from a time series of 1,600 seconds every half hour and quality controlled by FMI. The frequency at the spectral peak and the local water depth was used to calculate the phase speed, $c_p$, of the dominant waves. Then, the wave age, $c_p/U$, was calculated using the horizontal wind speed, $U$, as measured by the sonic anemometer at 10.4 m from the tower. Based on the wave age, three classes were defined: growing sea ($c_p/U < 0.8$), mixed sea ($0.8 \leq c_p/U < 1.2$) and swell ($1.2 \leq c_p/U$).

The wave age was only calculated for the open sea sector since the location of the buoy is not representative for winds from neither the Gotland sector nor the Östergarnsholm sector. Further, the land masses in these two sectors may have influenced the atmospheric properties as measured by the tower, which in turn complicates the analysis of the impact of the wave field.

Data from the buoy have previously been used to analyze the behavior of the wind profile and the turbulence properties of the atmospheric surface layer as measured by the mast at Östergarnsholm, see e.g. Semedo et al. (2009) and Mahrt et al. (2021).

### 3.5 Wind profile classification

All lidar wind profiles were classified into one of the following six classes: idealized, negative, LLmin, transition, weak LLJ or strong LLJ. As the definitions of LLJs vary in the literature, we follow the most recent recommendation by Aird et al.

(2021), applying both a fixed and a relative criterion for LLJ classification. Using hourly model data of wind profiles up to approximately 530 m height over Iowa (USA) during a period of 6 months (December 2007 to May 2008), Aird et al. (2021) concluded that defining LLJs based on only a fixed criterion compared to only using a relative criterion identified different LLJs 40% of the time. In terrain of low complexity (the Great Plains), the LLJ frequency was considerably higher using only the relative criterion. Aird et al. (2021) also showed that the definition affected the general characteristics of the LLJs: using

only the relative criterion, the statistics were biased towards LLJs with longer duration, lower core heights and lower core speeds, corresponding to a more stable atmospheric stratification with less turbulent kinetic energy. The opposite was true for the fixed criterion. For a robust result, Aird et al. (2021) recommended using a fixed criterion of either 2 m s$^{-1}$ or 2.5 m s$^{-1}$ in combination with the corresponding relative criterion of 20% or 25%.

In this study, a profile was classified as a strong LLJ if there was a well-pronounced local maximum in the wind profile

where the core speed was both at least 20% and at least 2 m s$^{-1}$ stronger than the weakest wind speed in the lidar profile both above and below the jet core. Thus, for core speeds below 10 m s$^{-1}$, 2 m s$^{-1}$ is the strongest criterion and for core speeds above 10 m s$^{-1}$, 20% is the strongest criterion. Similarly, for weak LLJs, the fixed and relative criteria were 1 m s$^{-1}$ and 10% respectively. Although 30 min data is used throughout this study, it is important to note that LLJ detection is sensitive to temporal averaging. In a test comparing the number of LLJs found in the time period 1 January 2018 – 24 June 2020, we

conclude that approximately 5% more strong LLJs (3% more weak LLJs) were found using 10 min data than in the 30 min data, analyzing comparable numbers.

Transition profiles were considered to be transitions between idealized profiles and LLJ profiles. They do display a local maximum in the profile, but fulfilling only criteria of 0.5 m s$^{-1}$ and 5% differences between the core speed and the lowest wind speed above and below the core. For profiles with a local low-level minimum in the profile (LLmin), the wind speed

above and below the 'core' had to be both at least 10% and at least 1 m s$^{-1}$ stronger than the speed at the local minimum. Negative profiles were defined as lidar profiles (28 – 300 m) where the wind speed decreased with height by at least 1 m s$^{-1}$ between the maximum and minimum wind speed and the profile was not fulfilling the criteria to be classified as an LLmin, a transition profile, a weak LLJ or a strong LLJ. It is important to note that for both the case of an LLmin and during negative profiles, there is a local maximum somewhere in the layer between the surface and the lowest measuring height of the lidar

data (28 m), which follows as a consequence of that the wind speed goes to zero at the surface.

All profiles that were not categorized as any of the non-idealized types described in the above, were classified as idealized profiles. Note that profiles with only a very slight negative shear and profiles with a very weak local minimum or maximum could be classified as idealized profiles. Also, note that only wind data from the lidar was used to classify the profiles (i.e. data from the meteorological mast was not included in the profile classification). Wind profile behavior above 300 m was not possible to assess using our data which was not considered a restriction as our main focus is to study the shape of the profiles in the height range most relevant for wind energy applications.

### 3.6 Classification of atmospheric stability

Atmospheric stability can be classified using many different approaches depending on the data at hand, e.g. using the Obukhov length (Obukhov, 1946), the flux, gradient or bulk Richardson numbers (e.g., Stull, 1988), or Pasquill classes (Pasquill, 1961). In this study, we chose the commonly used method (see e.g., Foken, 2006) of classifying the stability based on the stability parameter, $z/L$, where $z = 10.4$ m is the height of the measurements. The Obukhov length, $L$, was calculated as

$$L = -\frac{u_*^3 \theta_0}{\kappa g \overline{w'\theta_v'}} \tag{1}$$

where $\kappa = 0.40$ is the von Kármán constant, $g = 9.82$ m s$^{-2}$ is the gravitational constant, $\overline{w'\theta_v'}$ is the vertical flux of the virtual potential temperature (K m s$^{-1}$) and $u_*$ is the frictional velocity (m s$^{-1}$). Using standard notation for Reynolds decomposition, the prime denotes the turbulent fluctuations from the 30 min mean of the variable and the overbar denotes the mean of the product. The potential temperature, $\theta_0$, used in Eq. 1 was calculated as

$$\theta_0 = T \left(\frac{p_0}{p}\right)^{R_d/c_{pd}} \tag{2}$$

in which $T$ is the temperature measured by the CSAT3 sonic anemometer (K), $p_0$ is the reference pressure (1000 hPa), $p$ is the air pressure measured by the LI-7500 gas analyzer (hPa), $R_d$ is the gas constant for dry air (287.06 J kg$^{-1}$ K$^{-1}$) and $c_{pd}$ is the isobaric specific heat capacity for dry air (1004.71 J kg$^{-1}$ K$^{-1}$).

To obtain a frictional velocity for the total stress magnitude we used the definition given in Stull (1988),

$$u_* = \left(\overline{u'w'}^2 + \overline{v'w'}^2\right)^{1/4} \tag{3}$$

Using standard notation, $u$ is the horizontal wind speed in the dominant wind direction during the 30 min period of averaging, $v$ is the wind speed in the cross-wind direction during the averaging period and $w$ is the vertical wind speed. Thus, in Eq. 3, $\overline{u'w'}$ is the momentum flux in the along wind direction and $\overline{v'w'}$ is the cross-wind momentum flux, both measured in m$^2$ s$^{-2}$.

Using the stability parameter $z/L$, the local stability of the atmospheric surface layer could be classified. We used a five class system, unstable (U) when $z/L < -0.2$, weakly unstable (WU) when $-0.2 \leq z/L < -0.02$, near neutral (N) when $-0.02 \leq z/L < 0.02$, weakly stable (WS) when $0.02 \leq z/L < 0.2$ and stable (S) when $0.2 \leq z/L$. The thresholds were modified after the classification for offshore conditions presented by Sanz Rodrigo et al. (2015) – which in turn was based on Sorbjan and Grachev (2010) – and reduced from nine to five stability classes in order to obtain a sufficient amount of data in all classes (see supplement, Fig. S1) and to simplify the interpretation of the results.

## 3.7 Normalization of spectra

Similar to Smedman et al. (2004), we analyzed the turbulent $u$- and $w$-power spectra. The frequency, $n$, was normalized by the horizontal wind speed and the height of the measurements, $z = 10.4$ m, to obtain a normalized frequency, $f$, such that

$$f = \frac{nz}{U} \tag{4}$$

where $U$ is the average wind speed. Following the Kaimal et al. (1972) normalization (see also Sahlée et al. 2008) all $u$-spectra, $S_u(n)$, were normalized to coincide in the inertial subrange and allow for easier assessment of differences in the low frequency part of the spectra. The normalization was performed using the formula

$$\hat{S}_u(n) = \frac{nS_u(n)}{u_*^2 \phi_\varepsilon^{2/3}} \tag{5}$$

where $\phi_\varepsilon$ is the non-dimensional dissipation rate of energy

$$\phi_\varepsilon = \frac{\kappa z \varepsilon}{u_*^3} \tag{6}$$

The dissipation rate of turbulent kinetic energy, $\varepsilon$, was calculated as

$$\varepsilon = \frac{S_u(n_\varepsilon)^{3/2} 2\pi}{U \alpha^{3/2}} \tag{7}$$

where $n_\varepsilon$ is a selected frequency in the inertial subrange and $\alpha$ is the Kolmogorov constant for $u$. Note that combining Eqs.

5–7, the formula simplifies to

$$\hat{S}_u(n) = \frac{nS_u(n)}{S_u(n_\varepsilon)} \frac{\alpha U^{2/3}}{(\kappa z 2\pi)^{2/3}} \tag{8}$$

and thus, the normalization for a given spectra is a function of the average wind speed and the spectral value at $n_\varepsilon$. For $n_\varepsilon$ we chose the frequency 1.5 Hz and for $\alpha$ we used 0.52 (Högström, 1996). Using this representation of the $u$-power spectra, all spectra coincide in the inertial subrange, independent of stability, with a slope of $-2/3$ of the spectra in the inertial subrange,

if depicted in a log-log representation.

The $w$-power spectra was normalized by the variance of $w$,

$$\hat{S}_w(n) = \frac{nS_w(n)}{\sigma_w^2} \tag{9}$$

To compare the spectral values at a low frequency, the normalized frequency 0.01 was arbitrarily selected after visual inspection and with previous experience to predominantly represent a lower frequency than the spectral peak for the specific

measurement height used. Spectral values were then interpolated to this frequency from the neighbouring frequencies using linear regression in the log-log representation.

# 4 Results

## 4.1 General meteorological and oceanographic conditions

The general meteorological (temperature, wind speed, wind direction, stability of the atmospheric surface layer) and wave
conditions during the period of measurements are presented in Fig. 2. In Fig. 2(d), the data availability is plotted. Typically
the data availability – when data from all three instruments (sonic anemometer, lidar, buoy) were simultaneously available –
was approximately 50–80% per month. Note that all wind data from the sector 355°–45° was excluded from the analysis, as
mentioned in Sect. 3.1 and Sect. 3.2. For some months, such as April 2019, the data availability was very low (see Sect. 3.3 for
details) and the monthly statistics presented in Fig. 2 should be interpreted with care.

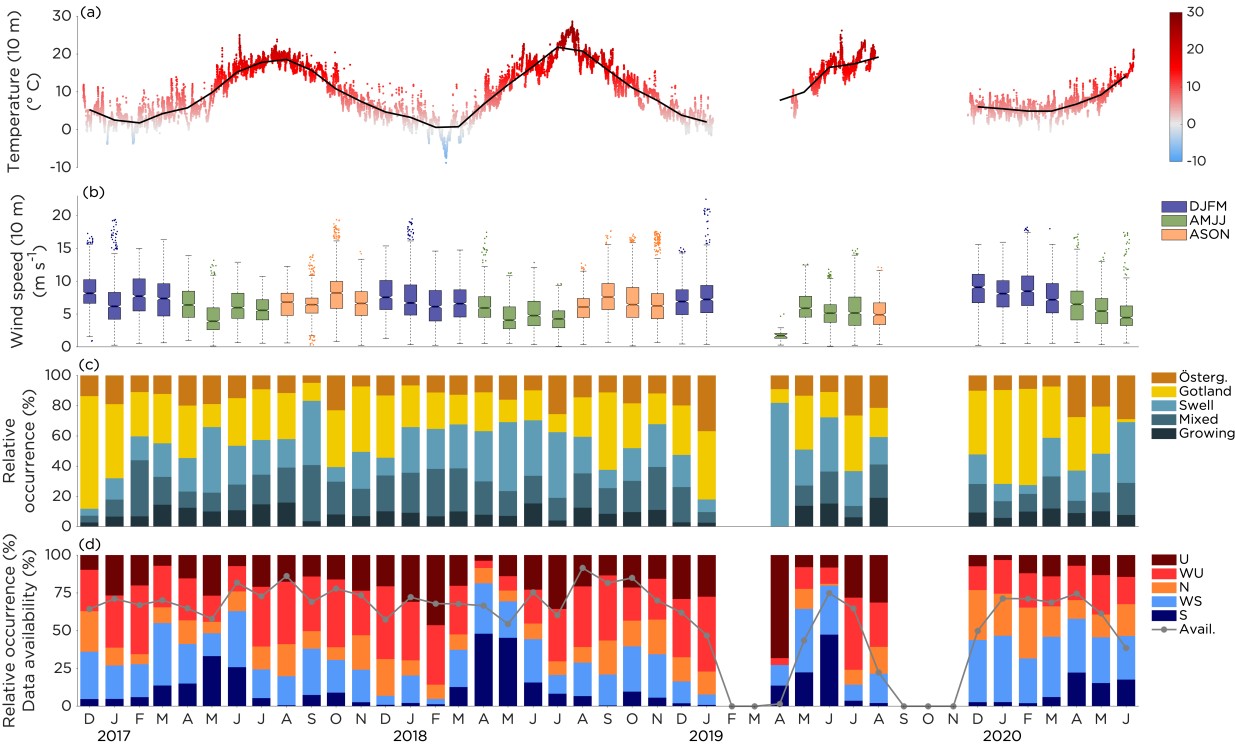

**Figure 2.** Overview of the (a) temperature (10 m), (b) wind speed (10 m), (c) wind direction (10 m) and wave age and (d) stability of the
atmospheric surface layer (see Sect. 3.6) during the period of measurements, 8 December 2016 to 24 June 2020. In (a) all 30 min average
temperatures are plotted together with the monthly mean (black line). In (b) the boxes are colored based on season (see Sect. 4.2). The line
in the boxes mark the median value, the bottom and top edges the 25th and 75th percentiles respectively. The dots indicate the outliers and
whiskers the most extreme wind speeds not considered outliers. The notches mark the 95% confidence interval of the median. In (c) the cases
when the wind was directed from the open sea sector were divided into growing sea, mixed sea and swell based on the wave age as described
in Sect. 3.4. In (d), the monthly data availability is plotted (grey line) in addition to the atmospheric stability.

Throughout the year, the temperature at 10 m height varied from −5°C in winter up to 25°C in summer, with a monthly
mean above 0°C for all months. No ice cover was reported in the close vicinity of Östergarnsholm in the winters during the
time period (SMHI, 2022). The monthly median wind speed at 10 m height was typically between 5 and 10 m s$^{-1}$, with
winter (DJFM) and fall (ASON) being the windier seasons and spring/summer (AMJJ) being less windy, especially in terms
of extremes. In Sect. 4.2 the reason for grouping the month into three seasons is presented.

For reference, the wind roses in Fig. 3 show the average wind speed and wind direction at Östergarnsholm for the three
seasons, which is in line with earlier studies at the site (see e.g. Svensson et al., 2019; Gutiérrez-Loza et al., 2019). The
dominant wind direction at the site was from S–SW, and thus winds from the open sea sector or from Gotland were most
common. When the wind was from the open sea sector, it was most often mixed waves or swell and less frequently growing
sea, Fig. 2(c). The waves were mostly directed from NE to SW, with waves coming from NE and SSW being more common
than waves directed from E (results not shown). In Fig. 3 the median wave age and the 25 and 75 percentiles of the wave age are
presented for the open sea sector, showing slightly higher wave age on average during AMJJ. This is attributed to the typically
lower wind speeds compared to the fall and winter seasons and is despite the fact that the phase speed of the dominant waves
is in general lower in AMJJ than in other seasons (see supplement, Fig. S2).

The stability of the atmospheric surface layer, Fig. 2(d), followed a yearly cycle where typically the unstable conditions
dominated during fall and winter while stable conditions were more common in spring and early summer. In total, for 18% of
the data the atmospheric stability was classified as unstable, for 30% as weakly unstable, 15% as neutral, 25% as weakly stable
and for 12% as stable.

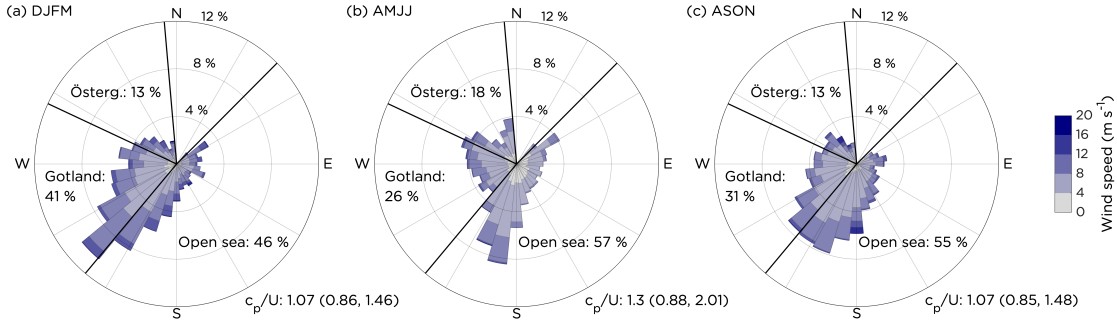

**Figure 3.** Wind roses showing the wind speed distributions at 10 m height for the three seasons (a) DJFM, (b) AMJJ and (c) ASON. Also
the relative occurrence of wind from the different sectors per season is given together with the median wave age for the open sea sector. The
25 and 75 percentiles of the wave age is given within the brackets.

## 4.2   Average profiles and monthly occurrence

The average wind speed and wind shear profiles from the lidar for the different types of profiles are presented in panels (a) and
(b) in Fig. 4. Negative profiles and profiles classified as LLmins typically occurred at lower wind speeds than other profiles
and, in turn, transition profiles and weak LLJs occurred at lower wind speeds than the idealized profiles. However, when a
strong LLJ was present, the core speed was typically stronger than the average wind speed for idealized profiles at the same

height. Earlier results from the site (Hallgren et al., 2020) indicate that although the LLJ core speed is typically in the range of 5–10 m s$^{-1}$, cases with core speeds exceeding 20 m s$^{-1}$ have been reported.

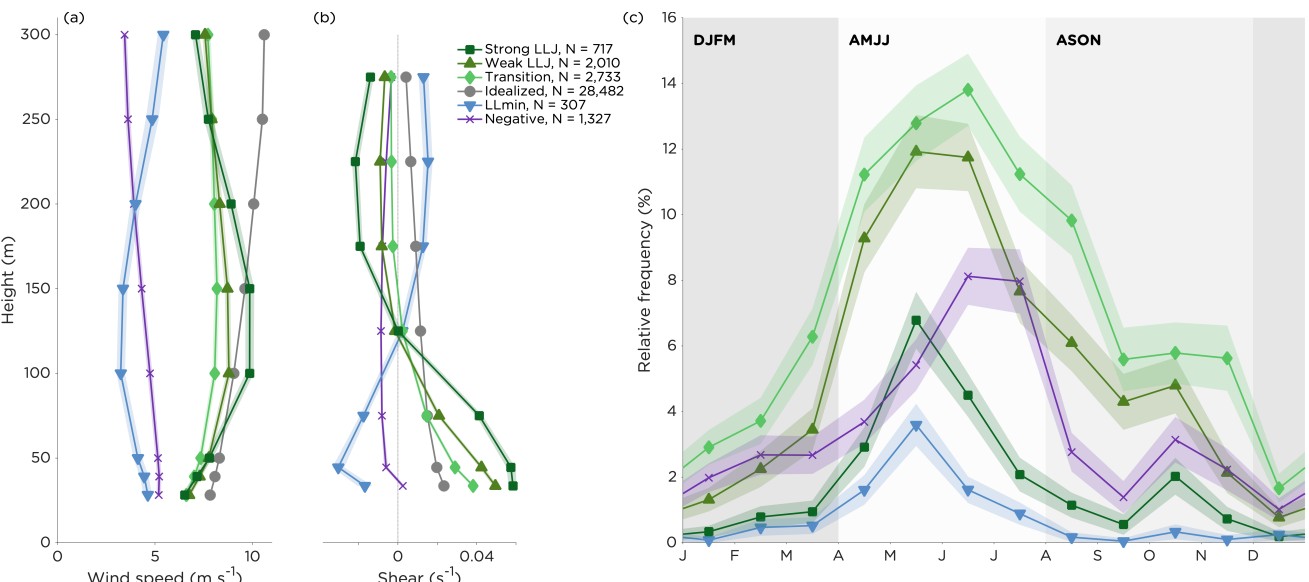

**Figure 4.** (a) Average vertical profiles of the wind speed and (b) wind shear for the different types of wind profiles. The total number of occurrences of the profiles is also given. In (c) the monthly average occurrence of the non-idealized profiles relative to all wind profiles is shown. The shaded areas around the profiles and in the relative occurrence plot indicate the 95% confidence interval of the mean.

The average shear profiles tell the same story as the average wind speed profiles, but from a different perspective. In the lowest layer of the profile, the average shear for the LLJ profiles was much stronger than the average shear for an idealized profile. By definition, the shear vanishes at the jet core and is negative above the core, even though the absolute value of the shear tend to decrease with height. Negative profiles had a shear that was relatively low all the time, although mostly on the negative side. Also, for LLmins, the absolute value of the shear was lower than for idealized profiles, except above the core

where the wind speed increased with height.

    The monthly average occurrences of the non-idealized profiles relative to all wind profiles are presented in Fig. 4(c). For all of these, there was a peak in the relative occurrence in the season April–July (AMJJ), reaching 35% in common, with a maximum value of approximately 40% in May and June. The high frequency of transition profiles, weak LLJs and strong LLJs in this season is related to the frequent occurrence of stable conditions at this time of year as relatively warm air heated

over surrounding land areas is advected over the Baltic Sea that is still relatively cold after the winter (Sect. 2.1, see also e.g. Svensson et al. 2016). During AMJJ the winds are in general weaker than during other seasons (Fig 2(b) and Fig. 3) and this, in combination with the generally higher values of the wave age (Fig. 3), is beneficial for formation of negative profiles and LLmins. However, the year to year variability of the relative occurrence of non-idealized profiles during AMJJ was large. For example, in May 2018, dominated by extended periods of atmospheric blocking with weak winds from the open sea sector,

non-idealized profiles occurred almost 60% of the time, while in May 2019 and May 2020, when the synoptic situation was more variable, the relative occurrence was around 30%. LLJs were more common in August–November (ASON) than in December–March (DJFM), but for the LLmins and negative profiles the difference was less clear.

Based on the seasonality seen in Fig. 4(c), we divide the months into three seasons to be used in the following analysis: winter (DJFM), spring/summer (AMJJ) and fall (ASON). No clear diurnal pattern could be seen in the occurrence of the different types of profiles during the different seasons, which is in line with results from earlier studies of the offshore LLJ (Hallgren et al., 2020; Aird et al., 2022).

The total occurrences of the different profiles are presented in the legend in Fig. 4(b) and, notably, negative profiles occurred as often as 4% of the time. Strong LLJs occurred 2% of the time and weak LLJs 6%. Transition profiles were rather common, occurring 8% of the time. LLmins were however rare, appearing approximately only 1% of the time. Please note that these numbers represent the record of the data as it is, and thus the summer season is slightly over-represented (since e.g. no data for February, March and September–November 2019 were available, see Sect. 3.3 and Fig. 2).

### 4.3 Occurrence in different wind speeds and wave ages

To assess when the different types of wind profiles appeared, distributions of 10 m wind speed and wave age for the three seasons are plotted in Fig. 5. The distributions are normalized for each type of profile. Both in DJFM and AMJJ it is clear that the peak of the distributions for the non-idealized profiles were shifted towards weaker wind conditions at 10 m compared to idealized profiles, see also Fig. 4(a). On the other hand, Fig. 4(a) also suggests that at heights relevant for wind power, LLJs occur in the range of wind speeds where the power curve typically is steep, implying that the power production could be very sensitive to the speed of the jet core. In ASON, the difference between the distributions of wind speeds for non-idealized profiles and idealized profiles was less pronounced, even though a larger share of the negative profiles were occurring in weak winds. Although rare, it is interesting to note that negative profiles sometimes occurred in stronger winds ($> 10$ m s$^{-1}$), primarily in ASON. Regarding the LLJs, it was very unlikely that they would appear if the wind speed surpassed 10 m s$^{-1}$ at 10 m height. As the wind speed decreases with height for the negative and LLmin profiles, most of these profiles occurred in wind speeds that were below typical cut-in wind speeds at standard hub heights for offshore wind turbines.

Panels (d), (e) and (f) in Fig. 5 answer the question: among the wind directions from the open sea sector, how many percent of a specific type of wind profile occurs in the different wave age classes (keeping the relative occurrence of the different types of profiles and of the wave age classes in mind) in a specific season? In terms of wave age, most of the negative profiles and LLmins occurred during swell conditions for all seasons. Strong LLJs were relatively uncommon in growing sea conditions, which is also the wave age class that was least frequently occurring (see also Fig. 2). Mixed sea and swell occurred by approximately the same frequency in DJFM and ASON. However, in AMJJ, swell was the dominant wave age class. Note that the wave age was only classified when the wind was directed from the open sea sector (see Sect. 3.4 and Fig. 2) and thus the percentages for the different wave age classes presented in Fig. 5 do not sum up to 100%.

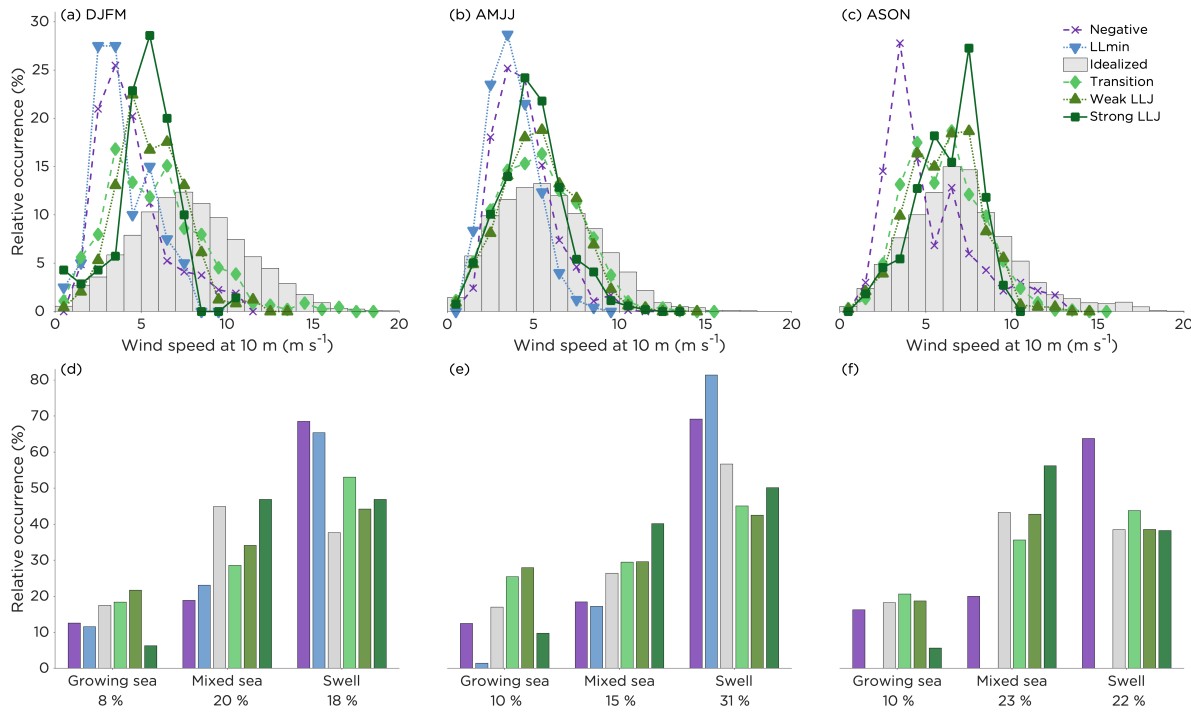

**Figure 5.** Seasonal distributions of (a), (b) and (c): wind speed at 10 m (all sectors); and (d), (e) and (f): wave age classes (only open sea sector) for non-idealized wind profiles compared to idealized profiles. The panels (a) and (d) represent DJFM, (b) and (e) represent AMJJ and (c) and (f) represent ASON. The relative occurrences of the different wave age classes in the different seasons are also given. Note that for each season and for each type of profile the relative occurrences add up to 100% combining the bars for the different sea states. A minimum of 20 occurrences per season was used to compile the statistics for each type of profile and thus LLmins are not plotted in ASON.

## 4.4 Occurrence in different wind directions and stabilities

In Fig. 6 the distribution of strong and weak LLJs and LLmins between different seasons is plotted, indicating in which wind speed (at 10 m height) and wind direction they occur, as well as in which atmospheric stability (also measured at 10 m height). Statistics for the different wind profiles regarding their features and the stability classes in which they appear are also presented for the three different wind direction sectors.

Comparing the polar scatter plots for strong LLJs in Fig. 6(b) and (c) with the wind roses in Fig. 3, it is clear that this type of wind profiles was over-represented when the winds were directed from the open sea sector during AMJJ and ASON (71 and 81% of all strong LLJs in the season, while in total winds from this sector appeared 57 and 55% of the time, respectively). The same was also true for weak LLJs, Fig. 6(e) and (f), with 74 and 76% relative occurrence in the open sea sector in AMJJ and ASON, respectively. However, during DJFM, a much larger share of the season's strong LLJs were from the Gotland sector (49%) compared to the other seasons (22% in AMJJ, 19% in ASON).

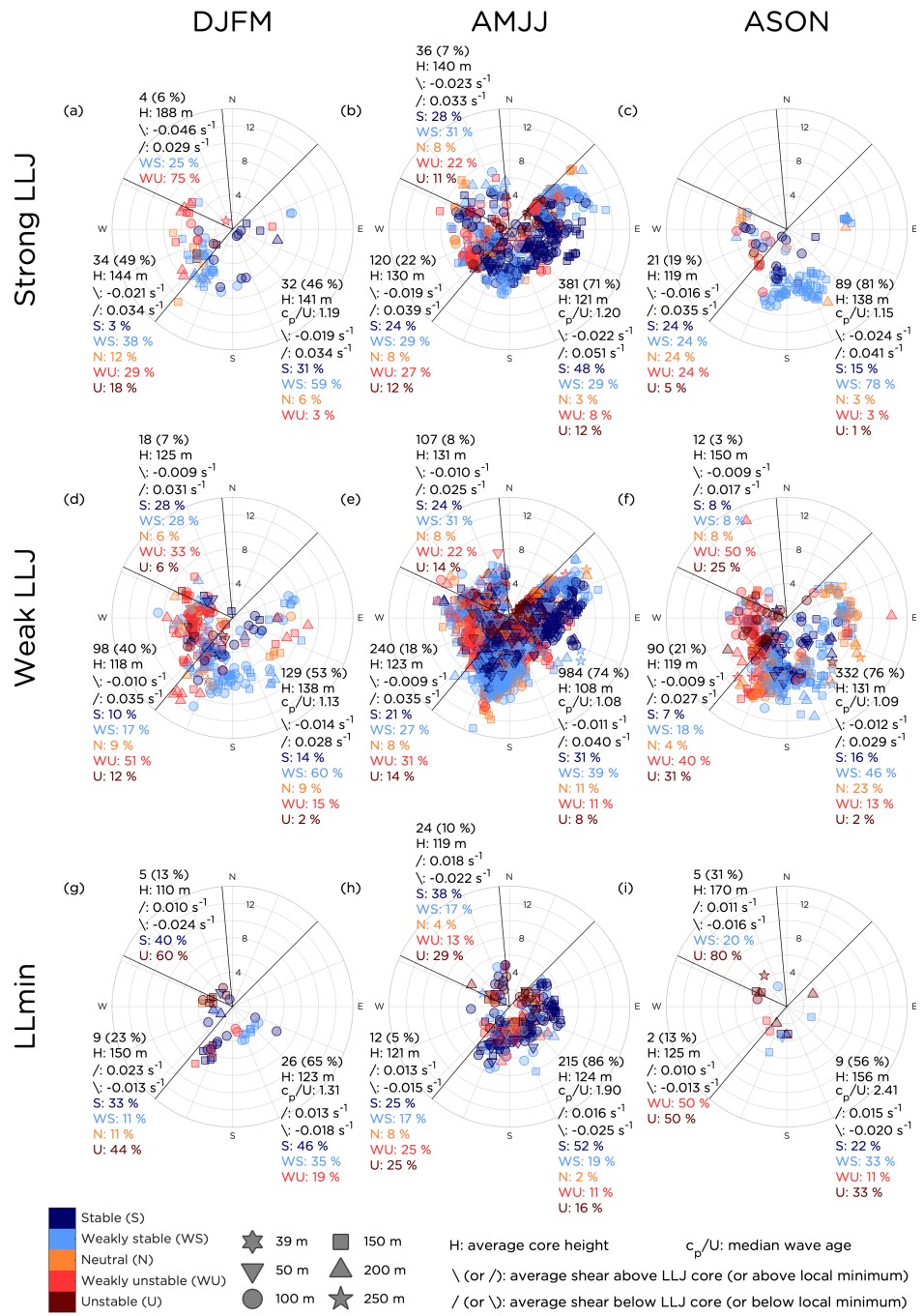

**Figure 6.** Overview of the seasonal occurrence of all profiles classified as (a), (b) and (c) strong LLJs; (d), (e) and (f) weak LLJs; and (g), (h) and (i) LLmins. The left column represent DJFM, the middle AMJJ, and the right ASON. The position in the polar diagram indicates the wind speed and wind direction at 10 m height, the color the stability of the atmospheric surface layer (Sect. 3.6) at the time of the occurrence of the profile. The different symbols indicate the height of the LLJ core or, for the LLmins, the height of the minimum. Statistics regarding the profiles are presented for each sector. Larger versions of the panels can be found in the supplement, Fig. S3–S11.

As a consequence of the height discretization in the lidar data, LLJs were limited to have cores located on the intermediate levels (the six levels between, and including, 39 and 250 m), as the wind speed by definition has to decrease compared to the levels above and below the core. Some differences in LLJ core height between sectors can be seen during AMJJ for both strong and weak LLJs, Fig. 6(b) and (e), primarily that the core height tended to be lower when the wind was directed from the open sea sector compared to sectors influenced by land surfaces. Comparing sector by sector for the different seasons, the LLJ core was in general located slightly higher up for strong LLJs than weak LLJs. Most strong LLJs appeared with a core height of 100 or 150 m (results not shown), averaging 128 m, weak LLJs averaging 118 m. It should be noted that very few LLJs (both strong and weak) were registered in winds from the Östergarnsholm sector both in DJFM and in ASON and thus the corresponding statistics should be interpreted with care.

As expected from theory, see Sect. 2.1, a vast majority of the LLJs appeared in stable stratification, at least when the wind was directed from the open sea sector. As an example, in AMJJ, 77% of the strong LLJs appeared in stable/weakly stable conditions, Fig. 6(b). However, for weak LLJs compared to strong LLJs, a growing share of the LLJs appeared in near-neutral or unstable/weakly unstable conditions. This can be seen comparing e.g. Fig. 6(c) and (f) where 93% of the strong LLJs in ASON (winds from open sea sector) occurred in stable or weakly stable stratification, while the corresponding number for weak LLJs was only 62%. In the statistics it can also be seen that, for all seasons, when the wind was directed from the Gotland or Östergarnsholm sectors, the LLJs also appeared more frequently in near-neutral or unstable/weakly unstable conditions compared to the open sea sector. The most prominent example of this being weak LLJs in ASON, Fig. 6(f), where 71% of the LLJs were occurring in unstable or weakly unstable stratification. It should also be noted that as LLJs often appear in events lasting for many hours and the statistics in Fig. 6 should be interpreted with this in mind as every 30 min wind profile classified as an LLJ is plotted. For example, the group of strong LLJs that appearing in ASON when the wind was directed from the E–ENE, Fig. 6(c), were from the same event lasting 4 h, although the height of the LLJ core and the stability changed slightly during the event.

In general, it can also be seen that for both weak and strong LLJs the value of the average shear below the jet core is higher than the absolute value of the average shear above the core. This result applies for all seasons and all sectors, except for strong LLJs in DJFM, Fig. 6(a), when the wind was directed from Östergarnsholm. However, in this case, the sample size is very small. The difference in shear below and above the LLJ core is also visualized in the average profiles plotted in Fig. 4(b). Comparing median wave age for when strong and weak LLJs were occurring in the open sea sector to the result for all data in each season (Fig. 3) it can be noted that the wave age was well within the range of the 25 and 75 percentiles and in all cases rather close to the median.

Profiles with an LLmin, Fig. 6(g), (h) and (i) also primarily appeared when the wind was directed from the open sea (65% in DJFM, 86% in AMJJ, 56% in ASON). Also, as for the LLJs, a vast majority of the LLmins appeared when the stratification was stable or weakly stable. However, in contrast to weak and strong LLJs, the LLmins typically occurred when the wave age was high, close to or exceeding the 75 percentiles presented in Fig. 3. According to the definition given in Sect. 3.4, the wave field is said to be dominated by swell when the wave age exceeds 1.2, and thus the median wave ages for the different seasons (1.31 in DJFM, 1.90 in AMJJ, 2.41 in ASON) indicate that LLmins predominantly occur in swell conditions. However, as the

number of observations of LLmins in DJFM and ASON was small, see panels (g) and (i) in Fig. 6, those results should be interpreted with care. The same applies for the Gotland and Östergarnsholm sectors in AMJJ, Fig. 6(h), with only 12 and 24 profiles displaying LLmins respectively. Regarding the average height of the local minimum in the wind profile for the LLmins (126 m), it was similar to the core height for the weak and strong LLJs, which is also suggested by Fig. 4(a). The wind shear profiles, Fig. 4(b) shows that on average the absolute value of the shear below the local minimum is higher than the average shear above, also expressed by the numbers in Fig. 6(g), (h) and (i).

Negative profiles (Fig. 7) predominantly occurred in AMJJ (62% of all negative profiles occurred in this season) and when the air was advected from the open sea sector. In AMJJ, 89% of the negative profiles occurred in the open sea sector, Fig. 7(b). Swell waves were common when negative profiles were appearing, with a median wave age exceeding the 75 percentile in both DJFM and ASON, comparing the numbers for $c_p/U$ in panels (a) and (c) in Fig. 7 with the corresponding numbers in panels (a) and (c) in Fig. 3. Interestingly, negative profiles primarily appeared when the atmospheric stability was unstable or weakly unstable. Under these conditions without influence from ocean waves we might initially expect that a higher degree of turbulent mixing would act to mix away anomalous profiles and reduce the probability of non-idealized wind profiles. However, in such stratification under influence from wind-following swell, which is a common situation for this site, large-eddy simulations have shown the possibility of a collapse of turbulence within the marine boundary layer (Sullivan et al., 2008; Nilsson et al., 2012) and the turbulence level will be typically low in such situations of low surface friction. Furthermore negative profiles extended to the surface have previously been observed with low-level wind maxima near the surface (e.g. Smedman et al., 2009; Nilsson et al., 2012), which should be expected as the wind speed very near the surface will approach zero (or some low value related to weak surface current). The general low wind speed conditions during negative profiles, see also Fig. 4(a), indicated relatively low turbulent mixing, despite the stability being unstable or weakly unstable, which is consistent to previous large-eddy simulation results. The average shear between 28 and 300 m during negative profiles ranged between $-7 \cdot 10^{-3}$ and $-4 \cdot 10^{-3}$ s$^{-1}$, to be compared with Fig. 4(b).

In general, the sectoral distributions of wind directions when non-idealized wind profiles occurred followed the 3.5 year climatology presented in Fig. 3, or the amount of data was too small to draw any conclusions. However, in the case of strong LLJs from the open sea sector, they were over-represented in winds from E–NE and under-represented in southerly winds during AMJJ (detailed results not shown).

### 4.5 Spectral analysis

The normalized longitudinal $u$-power spectra was interpolated to a fixed set of logarithmically spaced non-dimensional frequencies and then the median value of the spectra for each frequency and each type of profile was calculated. The results, divided into different stability classes and different sectors, are plotted in Fig. 8. The figure is accompanied by Table 1, summarizing the number of individual spectra (for each type of profile) that constitutes the statistics for each median spectra presented in Fig. 8 and for the boxplots in Fig. 9 and Fig. 10. In all stabilities and for all sectors, the idealized wind profile was the most common type of profile. However, in stable stratification and when the wind was directed from the open sea, the total amount

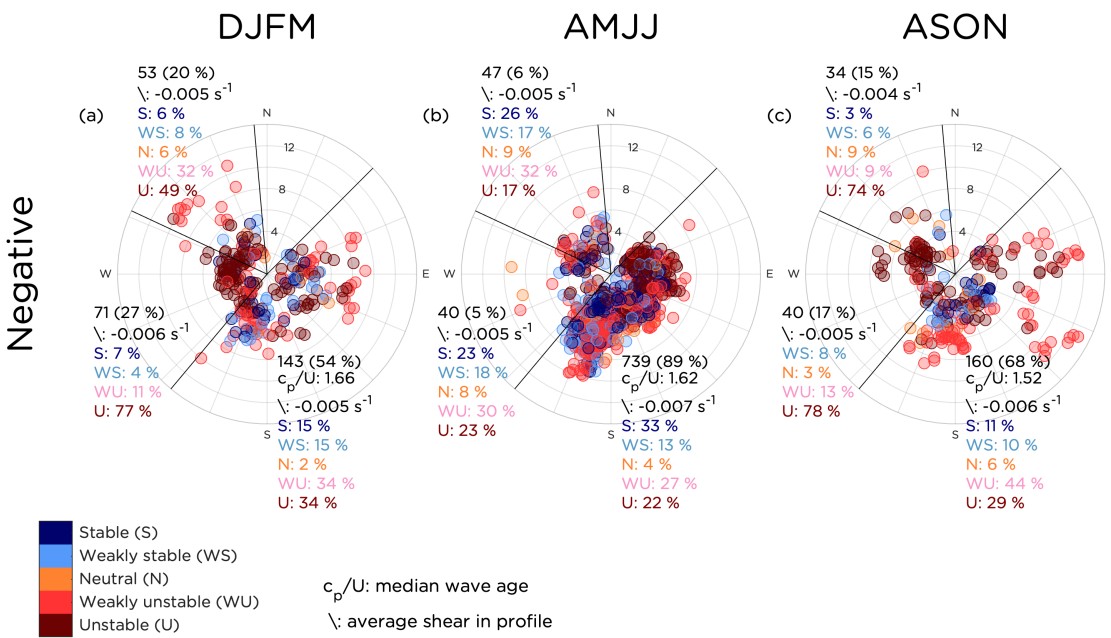

**Figure 7.** Overview of the seasonal occurrence for all profiles classified as negative profiles, presented in the same manner as in Fig. 6 with (a) corresponding to DJFM, (b) to AMJJ and (c) to ASON. Larger versions of the panels can be found in the supplement, Fig. S12–S14.

of non-idealized profiles (1,335) was somewhat greater than the amount of idealized profiles (1,079), which is remarkable keeping in mind that non-idealized profiles on average appeared approximately 20% of the time in total, see Fig. 4(c).

In Fig. 8, differences in the shapes of the spectral curves for the median values can be seen comparing different stabilities (for the same type of profile) or comparing different types of profiles (in the same stability class). The total spectral energy was lower in the weakly stable/stable conditions compared to more unstable conditions and the peak of the spectra was located at higher frequencies (smaller eddies) for stable conditions than in unstable conditions. Spectra for stable stratification displayed a clear spectral gap (especially for the open sea sector) at normalized frequencies of approximately 0.01 to 0.1. The increase of normalized spectral variances at frequencies lower than 0.01 indicate impacts from mesoscale effects and coherent structures such as dynamical density variations (e.g. Kelvin-Helmholtz instabilities) and atmospheric gravity waves which are common in stable stratification (Finnigan et al., 1984; Janssen and Komen, 1985; Yus-Díez et al., 2019). Within all stability classes, differences between the median spectra for different types of profiles could be observed in the low frequency range (large eddies). To analyze this further, the normalized frequency $nz/U = 0.01$ was selected following the methodology described in Sect. 3.7, and the spectral values at this frequency are presented in Fig. 9 (for longitudinal $u$-power spectra) and Fig. 10 (for vertical $w$-power spectra).

Starting by comparing the boxplots for transition profiles, weak LLJs and strong LLJs to the idealized profiles for the open sea sector in Fig. 9; as the notches do not overlap within any stability class there was a significant difference (5% significance level) in median spectral values, showing lower values for transition profiles and LLJs than for idealized profiles.

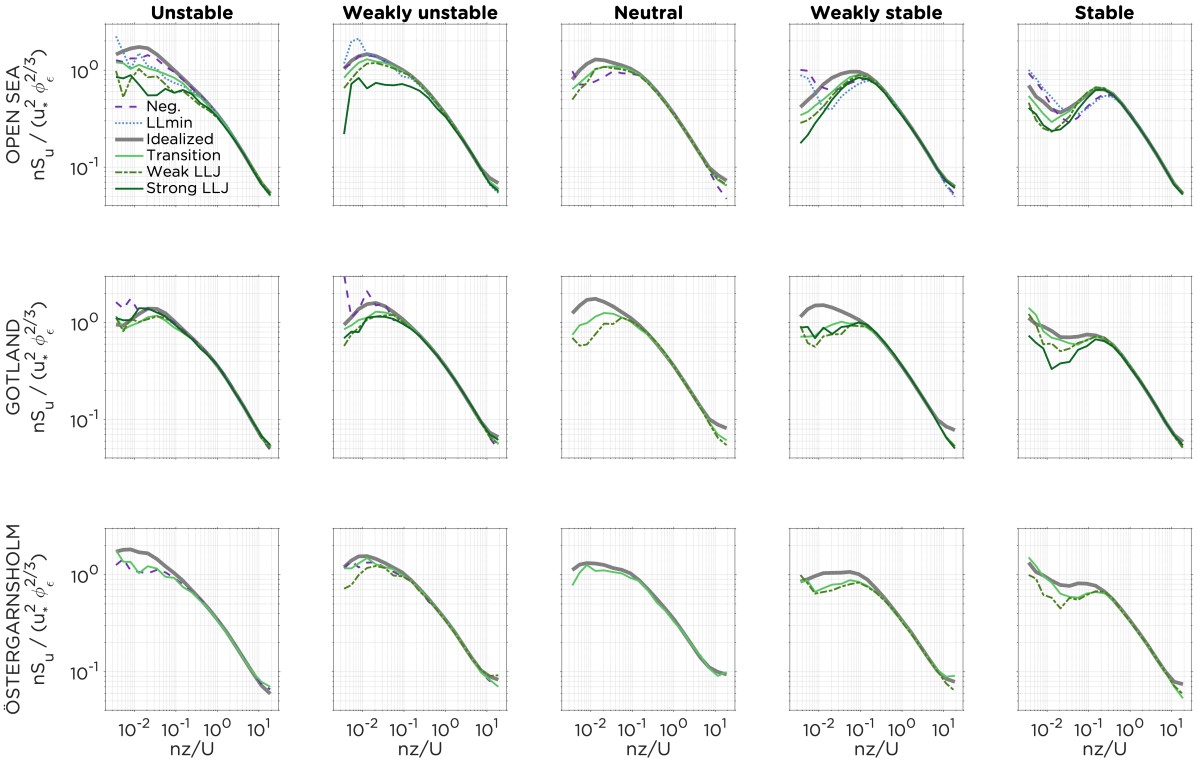

**Figure 8.** Median of the normalized turbulent $u$-power spectra plotted against normalized frequency for the different types of profiles occurring in the different wind direction sectors and under different atmospheric stabilities. A minimum of 20 occurrences of each type of profile per category was used as a limit to include the statistics for the profile, see Table 1.

The significance level comparing notches for the medians is based on an assumption of a normal distribution of the data for each box and that the number of samples are comparable (McGill et al., 1978). Thus, the significance of some of the results should be interpreted with great care as the sample size for idealized profiles in many cases outnumber the non-idealized profiles, see Table 1 for details.

For the Gotland sector, and similar to the results for the open sea sector, there was a difference in normalized spectral values at the selected low frequency between transition profiles and LLJs compared to idealized profiles in weakly unstable, near neutral and weakly stable conditions with the non-idealized profiles having lower spectral values. There was also a difference between the strong LLJs and the idealized profiles in stable stratification. Although even less data in the Östergarnsholm sector, there were indications of lower spectral values at the selected low frequency for wind profiles with an LLJ compared to idealized profiles. Similar results were also found if normalizing the $u$-power spectra with $\sigma_u^2$ instead of using the Kaimal et al. (1972) normalization (Sect. 3.7). A comparison of boxplots for the open sea sector created using the two types of normalization can be found in the supplement (Fig. S15). In the results for negative profiles and LLmins no clear patterns were detected in the

**Table 1.** Overview of the number of profiles (#) and the median stability ($z/L$) for the spectral values plotted in Fig. 8 and Fig. 10 for the different wind direction sectors and under different atmospheric stabilities. Note that $z/L$ values for profile types with less than 20 occurrences (per sector and stability) were omitted from being presented in the table. For each sector and each stability class, the values of $z/L$ that are higher than the corresponding value for idealized profiles are marked in **bold** (representing a shift towards less unstable or more stable conditions) and the lower values are marked in *italics* (representing a shift towards more unstable or less stable conditions).

| | | Unstable | | Weakly unstable | | Neutral | | Weakly stable | | Stable | |
| --- | --- | --- | --- | --- | --- | --- | --- | --- | --- | --- | --- |
| | | # | $z/L$ | # | $z/L$ | # | $z/L$ | # | $z/L$ | # | $z/L$ |
| Open sea | Strong LLJ | 46 | **-0.36** | 33 | **-0.04** | 18 | – | 200 | **0.09** | 205 | **0.47** |
| | Weak LLJ | 91 | *-0.76* | 173 | **-0.06** | 191 | 0.00 | 616 | **0.08** | 374 | *0.38* |
| | Transition | 166 | *-0.48* | 377 | **-0.06** | 281 | 0.00 | 692 | **0.08** | 345 | *0.38* |
| | Idealized | 2,679 | -0.46 | 4,310 | -0.08 | 2,332 | 0.00 | 3,158 | 0.06 | 1,079 | 0.43 |
| | LLmin | 37 | *-0.81* | 29 | *-0.09* | 5 | – | 53 | **0.12** | 126 | **0.70** |
| | Negative | 262 | **-0.44** | 319 | *-0.09* | 41 | 0.00 | 135 | **0.11** | 285 | **0.53** |
| Gotland | Strong LLJ | 21 | **-0.25** | 47 | *-0.09* | 19 | – | 53 | **0.10** | 35 | **0.37** |
| | Weak LLJ | 73 | *-0.38* | 160 | *-0.09* | 31 | **0.01** | 98 | **0.08** | 66 | *0.35* |
| | Transition | 147 | -0.36 | 210 | *-0.11* | 58 | *-0.01* | 113 | **0.08** | 86 | **0.43** |
| | Idealized | 1,805 | -0.36 | 2,843 | -0.08 | 1,617 | 0.00 | 3,096 | 0.06 | 874 | 0.36 |
| | LLmin | 8 | – | 4 | – | 2 | – | 3 | – | 6 | – |
| | Negative | 95 | *-0.51* | 25 | *-0.10* | 4 | – | 13 | – | 14 | – |
| Östergarnsh. | Strong LLJ | 4 | – | 11 | – | 3 | – | 12 | – | 10 | – |
| | Weak LLJ | 19 | – | 36 | *-0.10* | 11 | – | 39 | **0.09** | 32 | *0.47* |
| | Transition | 52 | *-0.45* | 68 | *-0.08* | 25 | **0.00** | 58 | **0.08** | 55 | *0.32* |
| | Idealized | 849 | -0.39 | 1,923 | -0.07 | 751 | -0.01 | 673 | 0.07 | 493 | 0.51 |
| | LLmin | 14 | – | 3 | – | 1 | – | 5 | – | 11 | – |
| | Negative | 59 | *-0.59* | 35 | *-0.13* | 10 | – | 14 | – | 16 | – |

spectral distributions comparing with distributions for idealized profiles. The median spectral value at the selected frequency varied between being higher or lower than the median spectral value for the idealized profiles.

Similar to Fig. 9, boxplots for extracted spectral values from the normalized $w$-power spectra for the same selected frequency $nz/U = 0.01$ are presented in Fig. 10. For the open sea sector, similar results as for $u$-power spectra could be identified with significantly lower values (5% significance level) for profiles with a local maximum in the profile compared to the idealized

profile. Also for the Gotland and Östergarnsholm sectors, the results for the $w$-power spectra resemble the results for the $u$-power spectra but the small sample size obstruct any conclusions. As the grouping of the data is the same as in Fig. 8 and Fig. 9, the numbers in Table 1 also represents the data presented in Fig. 10.

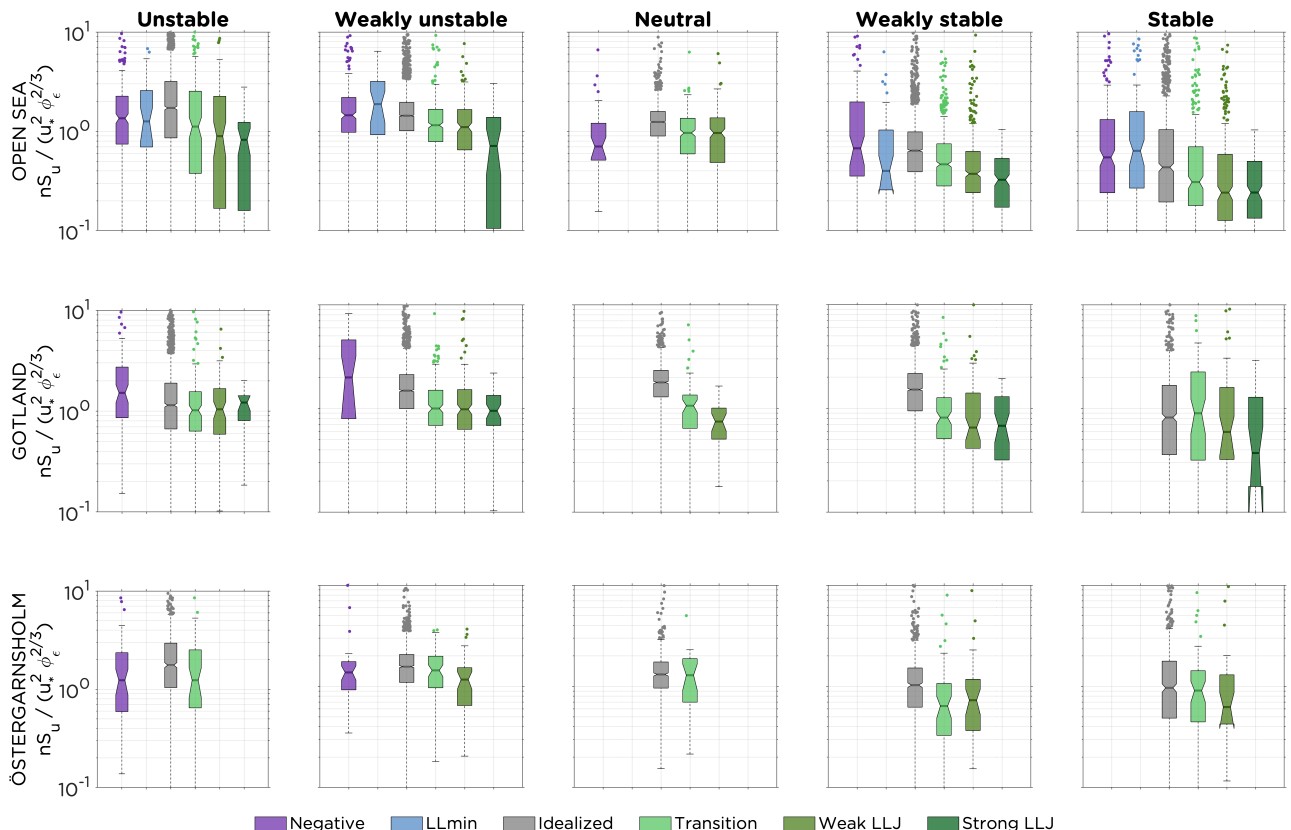

**Figure 9.** Distribution of normalized spectral values of $u$-power spectra for the different wind profile classes for the selected normalized frequency 0.01 (compare with Fig. 8). The data are categorized based on wind direction sector and stability of the atmospheric surface layer during the time of occurrence of the wind profile. The line in the boxes mark the median value, the bottom and top edges the 25th and 75th percentiles respectively. The dots indicate the outliers and the whiskers the most extreme spectral values not considered outliers. The notches mark the 95% confidence interval of the median.

## 5  Discussion

Several types of wind profiles, different from what here is classified as idealized profiles, are frequently occurring over the
Baltic Sea and presumably in any coastal area, e.g. St. Pé et al. 2018 and Møller et al. 2020, see also Svensson et al. 2016 for
a 14 year climatology of LLJ occurrence over the Baltic Sea and Kalverla et al. 2019a for a 10 year LLJ climatology over the
North Sea. Studying time series, it can be seen that transition profiles, weak LLJs and strong LLJs often appear clustered in
events, sometimes in close connection to the occurrence of negative profiles and LLmins. To assess wind power production
in the Baltic Sea area it is important to be aware of the frequent occurrence of these non-idealized wind profiles, how they
develop and their implications on the turbulence, shear stress on a wind turbine and consequences for wake behavior within
a wind farm. In the case of Östergarnsholm, non-idealized wind profiles occurred on average 20% of the time during the 3.5

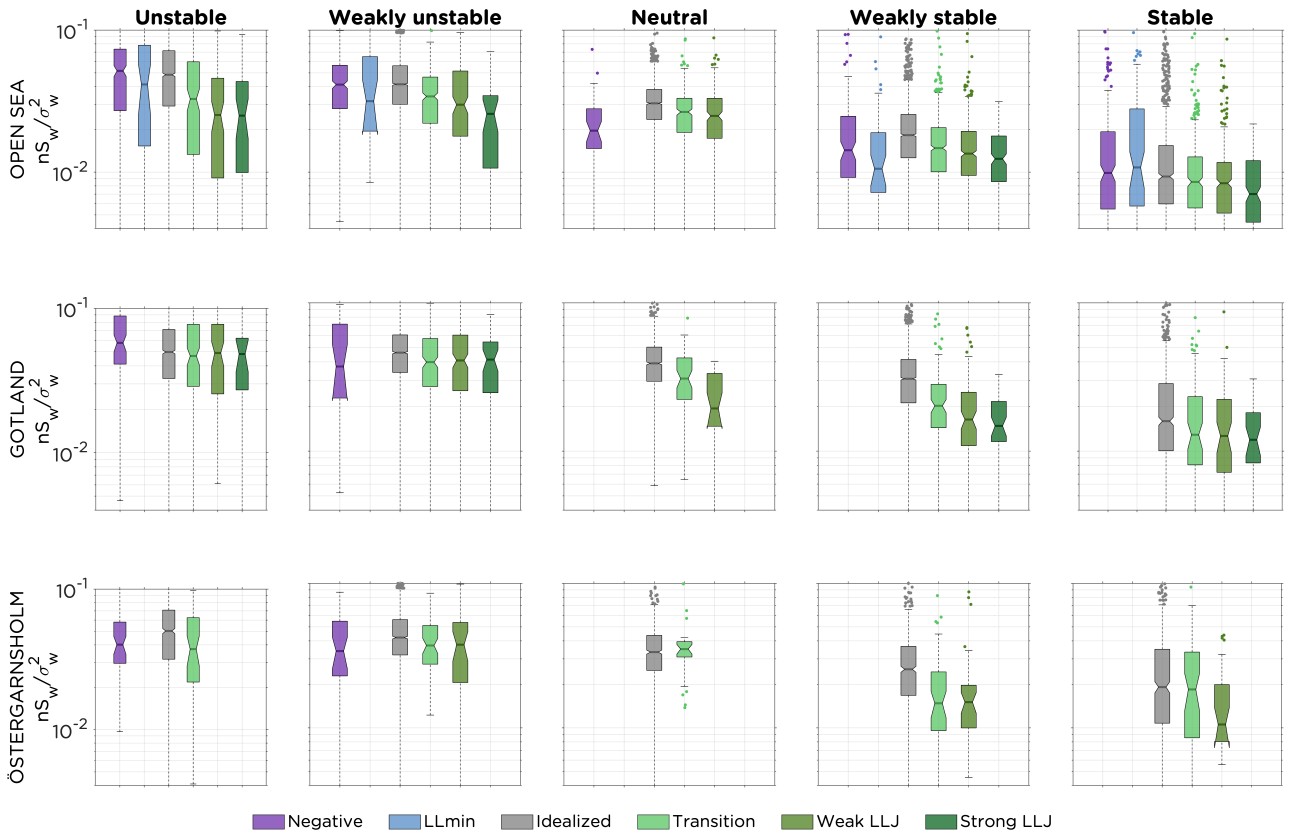

**Figure 10.** Distribution of normalized spectral values of $w$-power spectra for the different wind profile classes for the selected normalized frequency 0.01. The data are presented in the same manner as in Fig. 9.

years of measurements. In spring and summer (AMJJ), the stratification of the marine atmospheric surface layer was typically stable (see Fig. 2) as the water was still cold after the winter and warmer air was advected over the water surface, resulting in non-idealized profiles being even more frequent (see Sect. 2.1), occurring 35% of the time. Improving forecasts to be able to give information in advance about the occurrence of the wind profile type within the next few hours would provide a better basis for decision-making for wind power operators. Numerical weather prediction models, as well as other atmospheric models, struggle with resolving processes in the (marine) stable boundary layer (e.g., Kalverla et al., 2019b) and during swell (e.g., Wu et al., 2020) and this work strive to highlight – from an observational point of view – some of the processes occurring in the stable boundary layer, their turbulent properties and swell impact on wind profile behavior.

There are several interesting flow phenomena that may change the turbulence structure of the boundary layer, if the wind profile changes from an idealized logarithmic profile in a predominantly shear driven boundary layer where the wind shear smoothly decreases with height. A selection of these phenomena include shear sheltering (reduction of variance at low frequencies close to the surface and applicable only in stable stratification, see Sect. 2.2), dynamical instabilities (increased variances,

and potentially fluxes, at specific frequencies related to the shape of the wind profile), Holmboe instabilities (Holmboe, 1962;
Carpenter et al., 2012) (increased variances, and potentially fluxes, at specific frequencies related to the combined shape of
the wind and temperature profiles), modifications to transport of turbulence and/or pressure, and shear production close to the
surface caused by momentum input from swell waves (Semedo et al., 2009; Sullivan et al., 2008; Nilsson et al., 2012). Also,
modifications of the turbulence transport and shear production by horizontal gradients of wind and temperature can alter the
turbulent characteristics.

Smedman et al. (2004) found that during LLJs over the Baltic Sea, the energy in the low-frequency part of the turbulent
spectra was suppressed compared to corresponding cases without a low-level maximum in the wind profile. This was also earlier
discussed by Smedman et al. (1995), in which it was noted that gravity waves and large scale fluctuations were suppressed in
the case of LLJs with low core heights (in the 30–50 m layer). However, some studies (e.g., Duarte et al., 2012) have come to
opposing conclusions (Sect. 2.2), finding increased turbulence intensity at low frequencies during LLJs. In this study, using a
much longer data record than earlier work, we conclude that the median and average behavior of our data indicate a suppression
of large eddies in the layer below the LLJ core compared to the idealized case, at least when the wind was directed from the
open sea sector (Fig. 9 and Fig. 10). Also, the figures indicate that this suppression occurred in all stability classes, not only in
stable stratification which was the only stability class analyzed by Smedman et al. (2004). However, as mentioned above, there
are more possible explanations for this than only attributing it to shear sheltering.

As seen for the stable case in Fig. 8, there was a clear spectral gap separating the turbulent part of the spectra from larger
scale non-turbulent motions such as atmospheric waves. While it could be argued that these low frequencies should be filtered
out from the analysis (see e.g. Finnigan and Einaudi, 1981), we decided to perform the analysis in a similar manner as e.g.
Smedman et al. (2004), keeping all the turbulence data and analyse changes in the observed turbulence spectra, no matter
if atmospheric waves were present or not. It can also be noted in Fig. 8 that the increase in spectral energy at the lowest
frequencies was present not only for the case of an idealized wind profile, but also under non-idealized wind profile conditions.

Fig. 5 and Fig. 6(a)–(f), show that LLJs occurred in a wide range of wind speeds (at 10 m height). Note that the typical core
height of an LLJ is similar to the hub height of an offshore wind turbine (e.g., Gaertner et al., 2020) and that the core speed
is comparable to the average wind speed at that height, Fig. 4(a). Even in cases when the wind speed at hub height during an
LLJ matches the idealized conditions perfectly, it is expected that the power production will differ due to the shear profile of
the LLJ and thus changes in the rotor equivalent wind speed (REWS). The magnitude of the average shear in the layer from
the lowest measurement height in the lidar (28 m) to the height of the jet core is in general higher than the magnitude of the
shear from the jet core up to the highest measurement height (300 m). Since the rotor blades sweep heights ranging from 30
to 270 m in the case of the Gaertner et al. (2020) reference turbine, LLJs are expected to cause an uneven shear stress on the
rotor.

Since idealized profiles (occurring 80% of the time) cover a wide range of shear, a perfectly fair analysis would require
that only spectral values for LLJs and idealized profiles with similar shear in the lowest part of the profile were compared.
Using observational data and limiting to cases when the shear in the lower part of the profiles were similar and all governing
atmospheric and oceanographic conditions were the same, is however a hard restriction on the data. Thus, general practice in

the literature covering e.g. shear sheltering on atmospheric flow is to compare turbulent features for all LLJ profiles with all non-LLJ profiles (given the same stability close to the surface). In an attempt to see the effect on the turbulence spectra of idealized profiles with different shear, the rapid distortion technique was used (analytical solution as presented by Segalini and Arnqvist 2015) for a theoretical test case of constant shear and with constant stability throughout the boundary layer. It was concluded that the variance was lower in cases with stronger shear in the near-surface layer, however the effect disappeared when scaling the spectra with the variance.

Although spectra were classified according to their stability, the median values of $z/L$ given in Table 1 indicate that there was some spread within each stability class. Differences in the stability parameter $z/L$ within a class implies that the spectral values for cases with higher values of $z/L$ (more shifted towards stable stratification, marked in bold in the table) should have lower normalized variances in Fig. 9 and Fig. 10. Compared to the idealized profiles, the values for $z/L$ during strong LLJs were shifted slightly towards the more stable stratification for all stabilities in the open sea sector. However, the opposite was in many cases true for weak LLJs, and the strong signal of lower spectral values during both weak and strong LLJs visible in Fig. 9 and Fig. 10 can therefore be considered to be beyond the expected variation due to deviations in $z/L$. While the near-surface stratification at the height where the turbulence was measured is ruled out as the explanation for the difference in low frequency spectral density, there still exists an uncertainty with regards to the role of the stratification in the upper layer, above the jet core.

In contrast to LLJs, negative profiles typically occurred in unstable conditions and also at higher values of $c_p/U$ (Fig. 7). This suggests that the momentum flux was directed upwards (from the sea and the waves to the atmosphere), feeding energy into the lower part of the atmospheric boundary layer and thus also increasing the wind speed from below. In studies by e.g. Hanley and Belcher (2008) and Smedman et al. (2009), it was seen that a wind maxima can form at very low heights ($\sim 10$ m) during swell, and thus it is likely that the swell can contribute in creating a profile with negative shear (see e.g., Nilsson et al., 2012) as measured by the lidar (i.e. LLmins and negative profiles). Swell waves affect not only the wind speed profile, but also the wind direction and vertical profiles for turbulent kinetic energy and mixing length (see e.g. Wu et al. 2016 and Wu et al. 2020 for swell implications for offshore wind energy). The validity of using $z/L$ to describe the stability in situations with small (positive) momentum fluxes (i.e. in swell) could be questioned as these fluxes in reality could represent a turbulence regime that is very different from what we can attempt to describe with Monin-Obukhov similarity theory, see e.g. Smedman et al. (1995), Drennan et al. (1999) and Högström et al. (2013). Using Eq. 3 to describe the frictional velocity gives the magnitude of the stress, but information about the direction is lost. In total, positive longitudinal $\overline{u'w'}$ momentum fluxes occurred 3.5% of the time in the time period analyzed. For many of these cases the value of the momentum flux was so small that it could be considered to be within the uncertainty of the measurements if the flux should be considered positive or negative.

The lidar measurements used in this study were performed on eight height levels, most levels separated by a distance of 50 m. As LLJ cores can be of varying vertical extent, it is likely that this vertical resolution might not capture all LLJs. Sensitivity to height resolution (using modeling data) was discussed by Aird et al. (2021), concluding that coarser resolution resulted in lower average LLJ core height and core speed, shorter duration of LLJ events and lower spatio-temporal frequency. Also, as the measurement volume increases quadratically with height, the lidar is biased to capture sharper gradients and more localized

extreme points in the lower part of the boundary layer. Therefore, there will likely be the case that using continuous wave lidar data for any long term classification of wind profiles, cases with strong lower level shear and weak upper level shear will be overestimated. Further, averaging the data over 30 min also results in fewer LLJs than using higher temporal resolution, as was noted in Sect. 3.5.

When the fetch was affected by land surfaces (i.e. the wind was directed from the Gotland and Östergarnsholm sectors) the results regarding favorable conditions for formation of non-idealized profiles were less clear compared to when the wind was from the open sea sector (see Fig. 6 and Fig. 7), especially regarding stability. However, it is important to note that the stability was classified based on measurements of turbulence at 10 m height in the mast. As such, the classification might not fully represent the stability conditions when the air was transported over non-homogeneous terrain, as an internal boundary layer might be advected to the measurement site, affecting the wind profile at higher levels, or the measurements of turbulence in the mast. There are also other explanations for the unexpected result of LLJs occasionally occurring in unstable/weakly unstable stratification, Fig. 6(a)-(f), which is not in agreement with the general description presented in Sect. 2.1. For example, it is possible that the stability at 10 m can transition from stable to unstable while the 28–300 m wind profile remains unaffected for some time. Also for the open sea sector, it is not entirely correct to assume turbulence to be homogeneous within the footprint as e.g. the wave field or gradients in sea surface temperature might affect the fluxes. However, in a study by Högström et al. (2008) based on Östergarnsholm data it was shown that, during a five week campaign, measurements of momentum flux in the mast agreed well with measurements performed on an Air Sea Interaction Spar (ASIS) buoy located 4 km from the mast. No signal of differences in the measurements due to wave field heterogeneity was detected when the wind was directed from the sector 80°–210° (corresponding to the greater part of the open sea sector as defined in this study). Similar results were found for the sensible heat flux, although the scatter was larger. Thus, it can be assumed that open ocean conditions are measured by the mast at Östergarnsholm for winds from the open sea sector.

However, as was also noted in the study by Högström et al. (2008) the conclusions might not hold in cases of very light winds or during upwelling events, causing very sharp gradients in sea surface temperature. The effect of upwelling on wind speed and boundary layer height was assessed in a modeling study focusing on Gotland by Sproson and Sahlée (2014), and it was concluded that upwelling only have minor and very local effects on the atmospheric conditions due to the relatively strong winds in which upwelling typically occurs. It was noted that upwelling reduced the height of the boundary layer by up to 100 m, but only within 20 m from the upwelling area. For a climatology of upwelling in the Baltic Sea, we refer to e.g. Lehmann et al. (2012) and Zhang et al. (2022).

As a good understanding of the wind field in the lowest 300 m of the atmosphere in coastal regions is important for offshore wind power, this work has strived to deepen the knowledge about in which conditions non-idealized wind profiles occur in the coastal zone and how they might affect the turbulence in the boundary layer. Prior work has mainly been based on measurement campaigns or modeling studies, and the long data record provided in this study helps to bring more insights in these processes. For future work on alterations on the turbulence structure under influence of strong shear and in connection to different types of wind profiles, we suggest a combination of both modeling studies and extensive measurements.

Experimental evidence regarding the interplay between different effects that can alter the turbulent spectra has so far diverged, which calls for future studies on this topic. To explore the three-dimensional structure of different types of wind
profiles and processes related to their formation and evolution in different boundary layer stratifications and wind directions was not possible in this study. For this, model data is needed or a setup with a combination of horizontally and vertically scanning lidars. For the measurement site described in this study, additional measurements performed on buoys close to Östergarnsholm or at the east coast of Gotland would be beneficial in order to further investigate situations with internal-boundary layers.

When modeling the wind profiles in offshore conditions it is important to utilize a fully coupled model (Wu et al., 2020; Li et al., 2021), as feedback processes from the waves are important for the formation of e.g. negative profiles. This was confirmed from an observational perspective by the results in this study and becomes increasingly important in areas were swell is common (Hanley et al., 2010; Semedo et al., 2015). Using large-eddy simulations or fully resolving turbulence models would allow for an explicit analysis of the turbulent properties both above and below, as well as within, the LLJ core. We
suggest a modeling study that systematically investigates the impact on the spectral density in the case of boundary layers with strong low level shear and weak upper shear, ideally both including and excluding the effect of buoyancy. Turbulent proprieties above and below the LLJ core could also be assessed using approximate calculations of the momentum flux or turbulent kinetic energy from lidar measurements (Svensson et al., 2019; Thomasson, 2021), or at sites with tall meteorological masts, reaching up above the average LLJ core height and equipped with high frequency measurements at the top level. To measure wind
profiles and assess differences in atmospheric stability up to 300 m also e.g. radio soundings or drones could be used, or remote sensing devices, such as e.g. Radio Acoustic Sounding System (RASS).

Wind turbine wake behavior under different stabilities and for different wind speeds has been studied for both onshore (e.g. Iungo and Porté-Agel, 2014; Zhan et al., 2020) and offshore wind farms (e.g. Platis et al., 2022) but more research is needed to accurately describe the wind resource within wind parks and how power production can be optimized using wake steering.
The extent of the wake behind a wind turbine is likely to be very sensitive to LLJ conditions (e.g. Vollmer et al., 2017) as the potential to mix down momentum from above the jet core is greatly diminished.

Further studies on the impact from non-idealized profiles on the loads on a wind turbine and the total power production would be interesting as well as an assessment of methods to improve short-term forecasts with regard to these types of wind profiles. Clustering non-idealized profiles into events could give additional information about the synoptic and mesoscale conditions
and buoyancy mechanisms necessary for the formation of these type of wind profiles.

## 6 Summary and conclusions

Wind profiles with negative shear in at least one part of the profile between 28 and 300 m – weak and strong LLJs, transition profiles, LLmins and negative profiles – are frequently (∼20% of the time) occurring at Östergarnsholm, a coastal site in the Baltic Sea. From an offshore wind power perspective it is important to know when these profiles occur and how they might
affect the turbulent properties of the boundary layer, which in turn affects the loads on the wind turbine and the behavior of the

wake behind the turbine. Also, improving the understanding of physical processes in the boundary layer, especially in stable stratification, is important in order to improve the performance of numerical weather prediction models under these conditions.

By providing a systematic analysis of meteorological data and ocean wave conditions from over 3 years of measurements at Östergarnsholm, we concluded that:

- The non-idealized profiles exhibited a clear annual cycle; they were most common in AMJJ with a peak in May.

- The LLJs at Östergarnsholm were most frequently occurring in stable or weakly stable stratification but could appear in any stability. Also, LLmins primarily appeared in stable stratification while negative profiles were more frequent in unstable stratification.

- Most of the negative profiles and LLmins occurred during swell conditions.

- For all stability classes and during LLJs when the wind was directed from the open sea sector, lower normalized variances were found in the low frequency range of the spectra compared to idealized profiles. This follows the results found by Smedman et al. (2004). Further analyzes, considering processes both above and below the LLJ core, are needed to fully explain the cause for the observed change in turbulence properties.

- For LLmins and negative profiles there were no clear signals that these profiles altered the low frequency part of the $u$- and $w$-power spectra.

*Code availability.* The code used to generate the figures and tables can be acquired by contacting the first author (christoffer.hallgren@geo.uu.se)

*Data availability.* The data from the meteorological mast and the lidar are available from Erik Nilsson upon request. The data from the wave buoy are available from Heidi Pettersson upon request.

*Author contributions.* The project was conceptualized and administrated by CH and ES, with input from JA, EN, MS and SI. Funding acquisition was carried out by ES and SI. The methodology, programming, validation, formal analysis and visualization was performed by CH. CH also wrote most of the original draft, except Sect 3.2 (written by EN), Sect. 3.3 (written by JA) and Sect. 3.4 (written by HP). CH was supervised by ES and SI. AT performed the pilot study, supervised by CH. Data curation was performed by JA, EN and HP. All authors participated in reviewing and editing the manuscript.

*Competing interests.* The authors declare no conflict of interest. The funders had no role in the design of the study; in the collection, analyses, or interpretation of data; in the writing of the manuscript, or in the decision to publish the results.

*Acknowledgements.* This research was funded by the Energimyndigheten (Swedish Energy Agency) VindEl program, Grant Number 47054-1. The work forms part of the Swedish strategic research program StandUp for Wind. The ICOS station Östergarnsholm is funded by the Swedish Research Council and Uppsala University. The wave measurements were maintained with the help of the research infrastructure facilities provided by FINMARI (Finnish Marine Research Infrastructure network). We acknowledge Mr. Hannu Jokinen at the Finnish Meteorological Institute (FMI) for processing the wave buoy data. The authors would also like to thank Dr. Heiner Körnich at the Swedish Meteorological and Hydrological Institute (SMHI) for valuable comments during the course of the study.

735

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
