# Peer review of "Classification and properties of coastal wind profiles with negative gradients – an observational study"

_Wind Energy Science, 2021_

## Referee Comment (RC2)

Review of "Classification and Properties of Coastal Wind Profiles with Negative Gradients – An Observational Study"

General Comments:  This study presents multi-year datasets combined to address differences in the wind profile characteristics throughout the year as a function of sea state, stability, season, and turbulence (via spectral analysis of inertial subrange and the encompassing scales).  The different approaches toward conditioning the data made for an interesting read.  The results were well discussed; however, there are areas that need further explanation and details before formal acceptance can be made.  I would vote acceptance with minor reviews.  The specific comments below are mostly related to the science within the paper and areas where additional explanation is needed.  There were a lot of grammar/typo issues, and the reviewer likely missed many throughout the paper.  It is suggested that the authors use the examples below in the technical corrections section to improve other areas of the paper beginning at the results section and continuing to the end.

Specific Comments:

1. The authors investigate a multi-year dataset comprised of a tower, buoy, and a continuous wave lidar.  Given the distribution of wind profiles and the discussion of "normal" v. "non-normal" profiles, where the latter includes profile types resembling LLJs as well as negative gradients, then it is strongly recommended that the title is adjusted so that it would not seem that the authors are specifically focusing on wind profiles of negative gradients.  Afterall, that's only part of the story

2. Comment of usage of "normal" v. "non-normal": I would argue that using "normal" v. "non-normal" to describe wind profiles is related to the region of interest.  Certain regions, for example the great plains, experience LLJs during a fairly large proportion of days throughout the year.  The same can be said with the great barrier jet off the coast of California.  I would instead recommend the authors change "normal" to "idealized" and "non-normal" to "non-idealized" to describe departures from the ideal planetary boundary layer (PBL) model and the anticipated wind profile structure therein.

3. Your section 3 is entitled "Methods" and yet it goes over the sites and the measurements used in addition to methods.  I would change the title to reflect this.  Something like "Site, Measurements, and Methods"

4. Lines 161-164:  I don't quite understand this sentence fully.  Are you also saying the buoy can measure turbidity? No results related to ocean turbidity is reported in your study.  You do mention that wave properties have been evaluated against the turbulent state of the lower-atmosphere, but that is by making comparisons with the mast data. The sentence is confusing in other ways, too. Please improve your point here.

5. Line 218:  It's important to be sure that the linear interpolation is done over small time gaps.  You probably should make that clear in the paper.  Plus, I doubt that linear interpolation was done over large data gaps.  You probably want to comment on any impact this could have on the results and what your gap threshold is.  There are ways to approach this statistically as well if gaps are numerous and you have periods of similar turbulence intensity – with and without

data gaps – then you can compare those data and apply statistical randomization such as what is used when solving for the p-values.

6. Lines 345-347: It's hard to tell that the reason the wave age is higher is because of lower wind speeds. Clearly that could be born out the formulation that is used, but it is not shown clearly enough in Figure 3. The wind distribution over open seas is smaller during DJFM and the winds appear comparable when visually comparing the open sea sector distribution from DJFM with AMJJ. If you go ahead and multiply your wave age by U, then you should be able to compare the phase speeds between these periods to support this claim. Note your Figure 5 showing wind speed distributions and sea state. The AMJJ shows a much higher percentage of swell and more negative profiles, which weighs down the distribution toward lower wind speeds. The distribution, of course, can cannot easily be used to infer the wind rose distribution for open sector in figure 3.

7. Lines 406-409: The mechanisms behind the LLJ were not determined in this paper which makes determining why LLJ-type profiles existed during different stability regimes and for different sectors difficult. The Gotland sector has more LLJs for neutral-to-unstable conditions while the open sea has more LLJs during stable conditions. The site that this data was collected is small, so it is likely to impact the vertical structure of the LLJ over the island but not cause it. Another tricky part of this analysis is differentiating between what the local stability is over land compared to the stability over the ocean. The air being advected over the land site would also impact stability, too, in addition to surface heating/cooling, and this relative impact on stability is not easily characterized with the Monin-Obukhov framework.

8. Spectral Analysis: I know that the focus is mostly the inertial subrange and and frequency from which we depart from the inertial subrange into larger eddies. I agree with much of the discussion related to the spectral analysis, but I see that almost all "non-normal" profiles have lower energy containing eddies at the low frequency end. My guess is that one cause stems from dealing with much smaller sample sizes with the "non-normal" cases. It is interesting to note that only the negative profiles have examples where the largest energy containing eddies are larger than the "normal" profiles despite having the smallest wind shear and lightest wind conditions. Both Figure 8 and 9 show this. The production of turbulence is typically rooted in buoyancy or wind shear. Wind shear is small for the negative shear profiles so the thought is that buoyancy production should be responsible. However, the spectral energy is larger for stable conditions as well (Figure 9). Island meteorology and boundary layer internal advection is difficult. Can this be discussed or addressed in the paper?

9. Line 538-539: There are studies that look into turbine wake characteristics as a function of stability. Look at these such examples: Zhan et. al. (2020) and Iungo et. al. (2014)

Technical Corrections (T and G will denote typos (T) v. grammar (G); grammatical suggestions will be marked by S):

T1 (Line 8). Change "…that the the zone with strong shear during low-level…" to "…that the strong shear zone of low-level…"

S1 (Lines 15-16). Change "…This variation, the wind shear, plays a…" to "The vertical structure of the wind profile (i.e., wind shear and wind veer) plays an…." With this statement you could lump the two sentences mentioning wind shear and veer together

G1 (Line 22). Change "…grow with an…" to "…grow at an…"

G2 (Line 23). Change "…scenario and that…" to "…scenario since it is anticipated that…."

S2 (Line 27). Change "…Either replace the commas at the beginning and end of "a high latitude semi-enclosed" with hyphens or insert "which is" after "The Baltic Sea,"

G3 (Line 30). Insert "of" between "GW" and "offshore"

S3 (Lines 32-33). Recommend replacing ".. and expansion has to be performed with care." To "…and therefore expansion must be handled with care."

T2 (Line 35). Change to "increases". Forgot to add "s"

G4 (Lines 36-37). Change "…are prone to have wind profiles with partly negative gradients that can occur under certain meteorological and…" to "…are prone to having wind profiles with partly negative gradients under certain meteorological and…"

S4 (Line 40). Suggest changing "Note that also wind…" to "Note, also, that wind…"

G5 (Line 44). You use "both" to mention three additional factors. Both would apply to two additional factors. Perhaps change the following "…effects, both to assess the longevity of the turbines, the extension of the wake behind a single turbine and behind the park, and…" to effects; as well as to both to assess the longevity of the turbines, the extension of the wake behind a single turbine and behind the park, and…"

S5 (Line 53). Change "…what are the driving mechanisms for this." To "…what are the driving mechanisms that lead to turbulence production."

G6 (Lines 58-59). Change "In addition to this, not only the turbulent characteristics of the LLJs compared to normal profiles are analyzed…." to "In addition, not only are the turbulent characteristics of the LLJs compared to normal profiles…"

S6 (Lines 69-70). The sentence starting with "Already in 1957,…" is awkward. It just so happens that this is the year where this phenomenon was rigorously documented. I would consider starting this sentence with "One of the first proposed mechanisms related to the formation of…" You can fill in the rest.

S7 (Line 71). Instead of "During the evening and night,…" I would argue to use "During the evening transition,.."

S8 (Lines 74-75). The sentence beginning with "As a consequence…" is a bit of a mouthful. Perhaps make the following change: "…gradient force unbalanced, with a subsequent…." to "gradient force unbalanced. This imbalance subsequently leads to an acceleration of the wind: a process known as frictional decoupling."

G7 (Line 80). Change "As an effect…" to "As a result…" and change "compared to a water…" to "…compared to the water…"

S9 (Line 86). Recommend changing "at least if the swell and the wind direction are aligned which is the most studied case." to "…if the wind is approximately aligned with the swell direction"

T3/G8 (Lines 91-92). Change "…on theses matters, but for an introduction to uncommon wind profiles over the Baltic Sea and North Sea, we refer to Kettle (2014) and Moller et. al. (2020)." To "…these different wind profile types.  We refer to studies by Kettle (2014) and Moller et. al. (2020) for a description of less common wind profiles of this type."

G9 (Line 119).  Change "In a following…" to "In the following…."

T4 (Line 169).  Remove "is" after "as" and before "possible"

G10 (Line 185).  Do not need comma after "properly" and you remove "those" after "remove"

G11 (Line 210). I was confused with the wording here: "…site of especially swell conditions."  Were you trying to say "…site, especially during swell conditions."

G12 (Line 228).  Change "on" after "laser" and before "a" to "at"

G13 (Line 246).  I would change the sentence starting with "In this additional…" to something like "Application of additional quality controls led to the removal of 6.7% of the data."

G14 (Line 253).  Replace "on" between "depth" and "buoy" with "at"

General comment:  When introducing a variable, say, $C_p$, it is common to stick two commas between the variable.

G15 (Line 257). Replace "in" between "10.4 m" and "the" with "from"

G16 (Line 258).  Don't need a comma between "sector" and "since"

G17 (263).  Replace "on" between "mast" and "Ostergarnsholm" with "at".  I've seen this for multiple occasions.  I would make sure that this checked elsewhere.

---

## Author Comment (AC1)

Dear Referee #1,

Thank you for your detailed feedback and constructive comments on our manuscript! We have reviewed the manuscript according to your comments and made the following changes:

**Major remarks**

**1. My first main concern is that the paper does not bring what its introduction tends to promise. The introduction talks rather thoroughly about theories behind anomalous spectra in wind profiles like shear sheltering due to "separated" layers etc. However, in the analysis of the data set in the results section, the link to the theories is missing. The introduction raises the expectations that apart from the climatology of wind profiles that is promised and presented, it will also offer an explanation using these theories. Especially since the introduction concludes that there are a number of controversies about when anomalous spectra occur in which type of wind speed profiles. But that does not happen in the end. So I recommend to deepen the analysis by linking the anomalous profiles to the spectra and link them to the existing theories.**

This is a very important remark. We agree that the Theory section goes more in depth with the different possible explanations of what could cause changes in the turbulence spectra than the rest of the manuscript. Our aim with the Theory section was to give this broader background to the reader. The data set available offers limited possibilities of thoroughly investigating the physical mechanisms behind the behaviour of the spectra since the tower height of 30 m limits the information on turbulence structure to only the lower part of the boundary layer. Also, some of the theories are not possible to investigate further, such as the effect on the wind profile and spectra from horizontal gradients of wind and temperature. We have moved the part about the possible different flow phenomena that can alter the turbulence structure to the Discussion and also extended the discussion to better include the aspects that are brought up in the Theory section.

The Discussion was restructured and partly rewritten in order to better link to what is presented in the Theory section.

**2. Spectra: Concerning spectra I think the paper needs to justify more strongly why it needs the selected spectra to answer the research questions. I.e. why would for example wavelets not have given you a better of different (or similar answer). Secondly the manuscript needs to justify more whether the low frequency part of the spectrum in stable conditions (the ones you are interested in) are not affected or dominated by effects from atmospheric waves. The spectra show the traditional peak of turbulence, but then left of it there is a dip and then it increases again. This is typical for wave motions and their behaviour or impact (quantitative, time and height) on the wind speed profiles are likely different than by turbulence. See Enaudi and Finnigan papers from the 1980s how to separate waves from turbulence.**

Thank you for this interesting comment! We have added a paragraph in the Discussion section were the possible effect on the spectra from atmospheric waves is discussed. Looking at the spectra for the stable case in Fig. 8 it can be seen that the undulating shape of the spectra (the spectral gap) when the wind profile is non-idealized is similar to when the profile is idealized, but the spectral value of the local minimum is lower for the non-idealized cases, which is one of our main conclusions in the manuscript. It is reasonable to say that atmospheric waves could have a major impact on the shape of the spectra in stable conditions, but this could be the case both for the idealized case and for non-idealized cases.

Also, we agree that filtering out the atmospheric waves and only studying the turbulence in the low frequency part of the spectra separately is an interesting idea. However, as the Smedman et al. (2004) paper was the starting point for our study we decided to perform our analysis in a similar way, not filtering out only the turbulent part of the spectra and to use the Fourier transform when processing the turbulence data. With that said, it can also be noted that, although the Fourier transform is most commonly used in the literature we have come across, wavelet analysis was applied by e.g. Prabha et al. (2008) studying the influence of LLJs on canopy turbulence.

We do not expect any major differences in results using wavelets to determine the spectral shape. Minor differences should be expected depending on the choice of wavelet base, scale steps, and other settings, but our point is that the filtering process described in Sect. 3.2 of the manuscript is the most important step to

ensure statistically converged results also for the lower frequencies in the spectra. We maintain that in order to effectively investigate the cause of the behaviour, high quality turbulence measurements from higher heights are necessary, facilitating the study of cross spectral properties such as phase shifts between different heights.

We made the following addition to the Discussion (lines 585-590):

As seen for the stable case in Fig. 8, there was a clear spectral gap separating the turbulent part of the spectra from larger scale non-turbulent motions such as atmospheric waves. While it could be argued that these low frequencies should be filtered out from the analysis (see e.g. Finnigan and Einaudi 1981), we decided to perform the analysis in a similar manner as e.g. Smedman et al. (2004), keeping all the turbulence data and analyse changes in the observed turbulence spectra, no matter if atmospheric waves were present or not. It can also be noted in Fig. 8 that the increase in spectral energy at the lowest frequencies was present not only for the case of an idealized wind profile, but also under non-idealized wind profile conditions.

**3. I find the discussion section of the manuscript needs to be deepened. You correctly point out what are some of the weaker points of your analysis, or components that could not have been taken into account. But the discussion should go deeper about mainly a) what does these deficiencies concretely mean for the conclusions of the manuscript and b) how do the results compare to other studies, i.e. in other words please elaborate how this study has brought science a step forward. Some of the comments in the comments in the discussion section repeat what was already mentioned in the Introduction, but they remain disconnected from your own analysis and dataset.**

We have deepened the Discussion and tried to improve the connection between what is said in the Introduction and Theory sections and what the results show and how they can be interpreted. Further, we have tried to specify and argue which are the main conclusions from the study, why these results are important for future work and how the study has brought science a step forward.

**4. The paper discusses the wind speed profiles and turbulent spectra from a rather local point of view. To what extent can we assume the turbulence is homogeneous? I can imagine that sea surface temperature is spatially different across a gradient from north to south (and perhaps also east-west) which means the (in)stability is also spatially varying, which in turn affects the turbulence intensity. Hence one should be aware that part of the variances can have been advection from upwind. How would that affect your analysis. Can the analysis be extended by adding the SST (from ERA5 or any other data source) as a classifier?**

Thank you for this comment. We agree, saying that the turbulence is homogeneous within the footprint is not always correct. However, in a study by Högström et al. (2008) based on Östergarnsholm data, it was shown that during a five week campaign measurements of momentum flux in the mast agreed very well with measurements of momentum flux on an ASIS buoy located 4 km from the mast with no signal from wave field heterogeneity when the wind was directed from the sector 80°-210°. Similar results were found for the sensible heat flux, although the scatter was larger. Thus, it can be assumed that open ocean conditions are measured by the mast on Östergarnsholm for winds from the open sea sector (Rutgersson et al. 2020). However, as was also noted in the study by Högström et al. (2008) the conclusions might not hold in cases of very light winds or during upwelling events, causing large SST gradients, also supported by case study analysis in a MSc. thesis from 2010 (Román Cascón 2010) using flux towers, buoys and satellite images. It would be interesting to use high-resolution satellite observations for SST over long time periods to investigate non-local influence on the measurements, but this spatial analysis is outside the scope of our manuscript. We do not consider the effective spatial resolution of ERA5 or other reanalysis data sets enough to study the horizontal gradients of temperature and wind but more high-resolution coupled modeling studies may be needed for such an investigation. As we have noted in the Discussion, we suggest future modeling studies that could take these aspects into account. We also particularly note in the manuscript that the stability is measured locally in the mast (see e.g. lines 645-649 in the updated manuscript).

The following was added to the Discussion (lines 653-659):

However, in a study by Högström et al. (2008) based on Östergarnsholm data it was shown that, during a five week campaign, measurements of momentum flux in the mast agreed well with measurements performed on an Air Sea Interaction Spar (ASIS) buoy located 4 km from the mast. No signal of differences in the

measurements due to wave field heterogeneity was detected when the wind was directed from the sector 80°–210° (corresponding to the greater part of the open sea sector as defined in this study). Similar results were found for the sensible heat flux, although the scatter was larger. Thus, it can be assumed that open ocean conditions are measured by the mast at Östergarnsholm for winds from the open sea sector.

However, as was also noted in the study by Högström et al. (2008) the conclusions might not hold in cases of very light winds or during upwelling events, causing very sharp gradients in sea surface temperature. The effect of upwelling on wind speed and boundary layer height was assessed in a modeling study focusing on Gotland by Sproson and Sahlée (2014), and it was concluded that upwelling only have minor and very local effects on the atmospheric conditions due to the relatively strong winds in which upwelling typically occurs. It was noted that upwelling reduced the height of the boundary layer by up to 100 m, but only within 20 m from the upwelling area. For a climatology of upwelling in the Baltic Sea, we refer to e.g. Lehmann et al. (2012) and Zhang et al. (2022).

**5. Figures: The polar plots are nice but at the same time there are many datapoints plotted over each other with different colors, so it is a bit difficult to see what is the density of these categories of data points in the busiest polar plots. Are there any alternatives for plotting?**

We are aware of this problem and have tried to make adjustments to the polar plots so that the they should be easier to interpret. Also, we added larger versions of all panels in Fig. 6 and Fig. 7 in the supplement (Fig. S3–S14) that now accompanies the manuscript.

Minor remarks

Thank you for all the remarks that helped to improve the quality of our manuscript! If nothing else is stated below, we have directly implemented your suggestions.

**Ln 3: ....observational study...**

**Ln 5: non-normal wind profiles: unclear how it is defined. Do you mean wind profiles with negative shear only or do you mean something broader? Or just if the profile does not fulfill the logarithmic approach?**

On line 3 we state that "we classified non-idealized profiles as wind profiles having negative shear in at least one part of the lidar wind profile between 28 and 300 m". We clarified that it is this definition we use when referring to non-idealized wind profiles on line 5.

**Ln 51: shear sheltering. I think it is good to add a sentence or two explain what shear sheltering is. Just to make the manuscript attractive for a broader audience. Or refer forward that a more theoretical explanation will follow below.**

Thank you for this suggestion, we added a reference to the Theory section for the explanation of the term shear sheltering.

**Ln 115: the sensible heat flux. Do you mean surface sensible heat flux here? Or the sensible heat flux near the LLJ? Please clarify in the text.**

Smedman et al. (2004) analysed the sensible heat flux at the surface. We have clarified this in the text (line 118).

**Figure 1: Perhaps add a few labels where countries like Finland, Sweden, Estonia etc are located. This may help the non-European reader to understand better where you study was done.**

**Ln 155,156: This is a paragraph of one sentence, please merge with another paragraph.**

**Ln 157: with an undisturbed fetch of at least 150 km. Please add some words about the possibility that parts of the Baltic Sea can become frozen in winters. Did that happen in the years covering your dataset and within the footprint of the site (i.e. the 150 km you mention).**

We added the following sentence (lines 150-152): In winter, sea ice can cover the northern part of the Baltic Sea, bays and coastal areas, but during none of the winters in the period December 2016 to June 2020 the maximum ice extent affected the length of the fetch in the open sea sector (SMHI 2022).

Since before it is also noted in the Results on line 361 that no ice cover was reported in the close vicinity of Östergarnsholm during the time period of the measurements analysed.

**Ln 210: could you add which percent of the time the wind speed was below 3 and 1 m/s respectively?**

The percentages were calculated and added to the text which now reads (lines 208-212):

[...] it would imply a decrease of approximately 46% for 10 m winds below 3 m s$^{-1}$ and a 71% decrease of data availability for wind speed conditions below 1 m s$^{-1}$ (winds below 3 m s$^{-1}$ and below 1 m s$^{-1}$ occurring approximately 11% and 1% of the time respectively). This would have caused severe restrictions to the analysis at the Östergarnsholm site, especially during swell with winds below 3 m s$^{-1}$ (these wind speeds occurring approximately 26% of the time with swell).

**Ln 235: 30 minute averaging window. Can you elaborate a bit about what are the consequences of the relatively large averaging time for your results. If 5-min or 10-min averages would be used (so less averaging), one may catch more anomalous wind profiles than when one averages for 30-min. So are your results and conclusions conservative estimates?**

This is an important aspect of the analysis. We added this sentence (lines 235-237):

The relatively long time averaging window reduces the relative frequency of non-idealized wind profiles compared to if a shorter time frame would have been used, see Sect. 3.5 for further comments regarding the consequences of time averaging.

Since earlier, the following was included in the Discussion:

Further, temporal averaging results in fewer LLJs. Testing the sensitivity of temporal averaging, comparing the number of LLJs found in the time period 1 January 2018 – 24 June 2020, we conclude that approximately 5% more strong LLJs (3% more weak LLJs) were found using the 10 min data than in the 30 min data, analysing comparable numbers.

This text has now been partly rewritten and moved to Section 3.5 (lines 283-286) to simplify for the reader.

**Ln 266: This is a paragraph of one sentence, please merge with another paragraph.**

**Ln 268: Can you elaborate how much your results and conclusions are affected by the LLJ criterion? The manuscript is rather brief about it here.**

This is a good remark. We have elaborated on our definition of the LLJ by adding the following (lines 269-278):

As the definitions of LLJs vary in the literature, we follow the most recent recommendation by Aird et al. (2021), applying both a fixed and a relative criterion for LLJ classification. Using hourly model data of wind profiles up to approximately 530 m height over Iowa (USA) during a period of 6 months (December 2007 to May 2008), Aird et al. (2021) concluded that defining LLJs based on only a fixed criterion compared to only using a relative criterion identified different LLJs 40% of the time. In terrain of low complexity (the Great Plains), the LLJ frequency was considerably higher using only the relative criterion. Aird et al. (2021) also showed that the definition affected the general characteristics of the LLJs: using only the relative criterion, the statistics were biased towards LLJs with longer duration, lower core heights and lower core speeds, corresponding to a more stable atmospheric stratification with less turbulent kinetic energy. The opposite was true for the fixed criterion. For a robust result, Aird et al. (2021) recommended using a fixed criterion of either 2 m s$^{-1}$ or 2.5 m s$^{-1}$ in combination with the corresponding relative criterion of 20% or 25%.

**Ln 277: This is a paragraph of one sentence, please merge with another paragraph.**

**Ln 279: This is a paragraph of one sentence, please merge with another paragraph. Make more coherent please.**

Thank you for spotting this. The paragraphs have been merged.

**Ln 290: why does g deviate from the traditional value of 9.81?**

As the acceleration due to gravity ($g$) is dependent on latitude and altitude, we have used the value of 9.82 m s$^{-2}$ in the calculations (more precisely, the correct value of the acceleration due to gravity for Östergarnsholm is 9.81711 m s$^{-2}$).

**Section 3.6: The paper needs to elaborate more on how robust the findings are concerning their dependency on the atmospheric stability. The stability categorization is somehow arbitrary and other classification could have been chosen as well, i.e. in terms of Richardson number, or Pasquill classes and others. This need to be addressed in more depth. L becomes undefined when turbulence vanishes.**

We agree that all categorisations of atmospheric stability have their pros and cons. However, using the stability parameter $z/L$ is a standard approach in boundary-layer meteorology in general (see e.g. Foken 2006) and has been tested thoroughly. Since the tower only reaches 30 m, we expect gradient based measures (e.g. the Richardson number) to give the same results, since measurements are performed so close to the surface, which in practice also means that there will be enough turbulence to measure $L$. In a performance test of the Pasquill classes (Chapman 2017) compared to actual temperature lapse rates from the surface to 100 m layer, it was concluded that the scheme is biased towards neutral stratification. Although the classification performed well under stable conditions, unstable conditions were heavily underestimated. Further, using additional instrumentation to calculate e.g. the Pasquill classes risk to reduce the amount of data in the time series, since data from the sonic anemometer is anyhow needed to assess the turbulence. Also, as is stated in Sanz Rodrigo et al. (2015): "The sonic method however remains the only way of measuring local stability without the use of empirical functions or theoretical assumptions".

The following text was added to the beginning of Section 3.6 (lines 303-306):
Atmospheric stability can be classified using many different approaches depending on the data at hand, e.g. using the Obukhov length (Obukhov 1946), the flux, gradient or bulk Richardson numbers (e.g. Stull 1988), or Pasquill classes (Pasquill 1961). In this study, we chose the commonly used method (see e.g. Foken 2006) of classifying the stability based on the stability parameter, $z/L$, where $z = 10.4$ m is the height of the measurements.

Regarding the actual limits used for the stability classes, it was based on the classification presented in Sanz Rodrigo et al. (2015) but reduced from nine to five classes to simplify the interpretation of the results and to avoid classes with very few data points (corresponding to the very (un)stable and extremely (un)stable classes used by Sanz Rodrigo et al. 2015). In the histogram in Fig. AC1.1 below (also included in the supplement, Fig. S1), the distribution of $z/L$ is plotted using a bin size of 0.01. The results are similar to those presented for coastal sites in other studies, e.g. Kelly and Gryning (2010).

The last sentence in Sect. 3.6 now reads (lines 323-326):

The thresholds were modified after the classification for offshore conditions presented by Sanz Rodrigo et al. (2015) – which in turn was based on Sorbjan and Grachev (2010) – and reduced from nine to five stability classes in order to obtain a sufficient amount of data in all classes (see supplement, Fig. S1) and to simplify the interpretation of the results.

[Figure]

Figure AC1.1: Histogram of the stability parameter, $z/L$, divided into the five different stability classes used in our manuscript. A bin size of 0.01 is used.

**Equation 5: Why is the Su spectrum not scaled with the variance of u? I.e. similar as for Sw. The paper needs to discuss why this normalization is suitable for the goal you want to achieve with the study. I would say the Su scaling as done now is more sensitive to measurement uncertainties than just scaling with $\sigma_u^2$.**

The reason for scaling the Su spectra according to the method described in Kaimal et al. (1972) (see also Sahlée et al. 2008) relates to our aim of investigating the low frequency part of the spectra. Using the given normalization, spectra collapse in the high frequency part (the inertial subrange) and differences in the low frequency part is easier to detect in the median spectra (Fig. 8). The sensitivity to measurement uncertainties are important for both normalization techniques but not for our main conclusions. To test the impact on the selection of normalization on the conclusions we have implemented the $\sigma_u^2$ normalization and compared the box plots for the open sea sector (Fig. AC1.2). Although the numbers differ, the general conclusion that LLJs display lower spectral power energy at the selected low frequency $nz/U = 0.01$ still holds.

[Figure]

Figure AC1.2: Comparison of box plots for the open sea sector for the two normalizations used for u-power spectra.

We have motivated our choice of normalization a bit further in Sect. 3.7 which now reads (lines 328-345):

Similar to Smedman et al. (2004), we analyzed the turbulent $u$- and $w$-power spectra. The frequency, $n$, was normalized by the horizontal wind speed and the height of the measurements, $z = 10.4$ m, to obtain a normalized frequency, $f$, such that

$$f = \frac{nz}{U} \tag{1}$$

where $U$ is the average wind speed. Following the Kaimal et al. (1972) normalization (see also Sahlée et al. 2008) all $u$-spectra, $S_u(n)$, were normalized to coincide in the inertial subrange and allow for easier assessment of differences in the low frequency part of the spectra. The normalization was performed using the formula

$$\hat{S}_u(n) = \frac{nS_u(n)}{u_*^2 \phi_\varepsilon^{2/3}} \tag{2}$$

where $\phi_\varepsilon$ is the non-dimensional dissipation rate of energy

$$\phi_\varepsilon = \frac{\kappa z \varepsilon}{u_*^3} \tag{3}$$

The dissipation rate of turbulent kinetic energy, $\varepsilon$, was calculated as

$$\varepsilon = \frac{S_u\left(n_\varepsilon\right)^{3/2} 2\pi}{U \alpha^{3/2}} \tag{4}$$

where $n_\varepsilon$ is a selected frequency in the inertial subrange and $\alpha$ is the Kolmogorov constant for $u$. Note that combining Eqs. 2–4, the formula simplifies to

$$\hat{S}_u(n) = \frac{nS_u(n)}{S_u\left(n_\varepsilon\right)} \frac{\alpha U^{2/3}}{\left(\kappa z 2\pi\right)^{2/3}} \tag{5}$$

and thus, the normalization for a given spectra is a function of the average wind speed and the spectral value at $n_\varepsilon$. For $n_\varepsilon$ we chose the frequency 1.5 Hz and for $\alpha$ we used 0.52 (Högström 1996). Using this representation of the $u$-power spectra, all spectra coincide in the inertial subrange, independent of stability, with a slope of $-2/3$ of the spectra in the inertial subrange, if depicted in a log-log representation.

We also added the following to the Results, Sect. 4.5 (lines 537-539):

Similar results were also found if normalizing the $u$-power spectra with $\sigma_u^2$ instead of using the Kaimal et al. (1972) normalization (Sect. 3.7). A comparison of boxplots for the open sea sector created using the two types of normalization can be found in the supplement (Fig. S15).

**Ln 320: This is a paragraph of one sentence, please merge with another paragraph.**

**Figure 3: the description of this graph does not explain whether and how the wind roses or climatology are consistent with earlier studies over the site or over the Baltic. Please bring in context.**

We have clarified the text regarding the wind roses (lines 365-366):

For reference, the wind roses in Fig. 3 show the average wind speed and wind direction at Östergarnsholm for the three seasons, which is in line with earlier studies at the site (see e.g. Svensson et al. 2019; Gutiérrez-Loza et al. 2019).

**Figure 4: caption: panel c does not show the normal type, while the caption suggests it is present. Please adjust the caption.**

The relevant part of the caption states that only non-idealized profiles are plotted in panel (c): In (c) the monthly average occurrence of the non-idealized profiles relative to all wind profiles is shown.

Thus, we did not make any change in the caption regarding this.

**Ln 358: goes to zero -> "vanish"**

**Ln 363: with a peak value of approximately 40% in May and June. Please add some words about the physical reasons why this peaks in May-June. I.e. interpret the climatological values.**

We have added the physical explanation for this. The part of the paragraph now reads (lines 391-398):

The monthly average occurrences of the non-idealized profiles relative to all wind profiles are presented in Fig. 4(c). For all of these, there was a peak in the relative occurrence in the season April–July (AMJJ), reaching 35% in common, with a maximum value of approximately 40% in May and June. The high frequency of transition profiles, weak LLJs and strong LLJs in this season is related to the frequent occurrence of stable conditions at this time of year as relatively warm air heated over surrounding land areas is advected over the Baltic Sea that is still relatively cold after the winter (see e.g. Svensson et al., 2016). During AMJJ the winds are in general weaker than during other seasons (see Fig 2(b) and Fig. 3) and this, in combination with the generally higher values of the wave age (Fig. 3), is beneficial for formation of negative profiles and LLmins.

**Figure 6: Please extend the legend and make explicit what H, S, WS, ...U represent. It can be helpful to add the season labelling at the top of the rows of figures and the LLJ strength along the rows, such that the reader does not need to read the figure caption 5 times to unravel what is in all these plots.**

Thank you for this comment. We have implemented this according to you suggestion.

**Ln 403-404: naturally, the higher the LLJ core is located, the higher the core speed. I would say this is not so natural. If you look in the Van de Wiel paper that you refer to in the Introduction you will see that the largest amplitude of the inertial oscillation occurs near the ground and as such the highest LLJ magnitude is expected not to be at higher levels. Please revise.**

You are completely right, thank you for spotting this. We have revised the sentence and removed the word "naturally".

**Section 4.4 as a whole: I recommend to make this section more quantitative in its discussion. The reader now has to do the analysis of figure 6 by her/himself and this may introduce different conclusions than what the authors intend to say.**

We have extended Sect. 4.4 to explain and interpret in more detail the results presented in Fig. 6 and Fig. 7.

**Ln 419: Negative profiles (Fig. 7) predominantly occurred when the air was advected from the open sea sector and the conditions were unstable or weakly unstable. Can you provide a physical explanation for this. Exactly in unstable conditions I would expect that the anomalous profiles would be mixed away by rather rigorous turbulence. Did you mean the negative profiles are related to the swell?**

We agree, this sentence was misleading in the original manuscript. Thank you for spotting this. We have tried to clarify by writing (lines 484-498):

Negative profiles (Fig. 7) predominantly occurred in AMJJ (62% of all negative profiles occurred in this season) and when the air was advected from the open sea sector. In AMJJ, 89% of the negative profiles occurred in the open sea sector, Fig. 7(b). Swell waves were common when negative profiles were appearing, with a median wave age exceeding the 75 percentile in both DJFM and ASON, comparing the numbers for $c_p/U$ in panels (a) and (c) in Fig. 7 with the corresponding numbers in panels (a) and (c) in Fig. 3. Interestingly, negative profiles primarily appeared when the atmospheric stability was unstable or weakly unstable. Under these conditions without influence from ocean waves we might initially expect that a higher degree of turbulent mixing would act to mix away anomalous profiles and reduce the probability of non-idealized wind profiles. However, in such stratification under influence from wind-following swell, which is a common situation for this site, large-eddy simulations have shown the possibility of a collapse of turbulence within the marine boundary layer (Sullivan et al. 2008, Nilsson et al. 2012) and the turbulence level will be typically low in such situations of low surface friction. Furthermore negative profiles extended to the surface have previously been observed with low-level wind maxima near the surface (e.g. Smedman et al.

2009, Nilsson et al. 2012), which should be expected as the wind speed very near the surface will approach zero (or some low value related to weak surface current). The general low wind speed conditions during negative profiles, see also Fig. 4(a), indicated relatively low turbulent mixing, despite the stability being unstable or weakly unstable, which is consistent to previous large-eddy simulation results.

**Ln 420: local stratification. This is confusing for the reader. How do you define stability in this study? Is there also a non-local stratification or spatial evaluation of the stratification involved? I thought it was just based on the measured L on the site.**

You are correct, only the local stratification as measured in the mast is used. We have clarified this and changed the text (see reply to previous comment).

**Ln 424: it can be seen. Please avoid passive sentences.**

The paragraph has been rewritten (lines 500-504):

In general, the sectoral distributions of wind directions when non-idealized wind profiles occurred followed the 3.5 year climatology presented in Fig. 3, or the amount of data was too small to draw any conclusions. However, in the case of strong LLJs from the open sea sector, they were over-represented in winds from E–NE and under-represented in southerly winds during AMJJ (detailed results not shown).

**Ln 435, 436: is the difference systematic, i.e. can you support evidence with results from a statistical test?**

Our main point here is that the number of non-idealized profiles when the wind is directed from the open sea and during stable conditions was remarkably high and even if the total number of non-idealized profiles under these circumstances would have been slightly lower than the number of idealized profiles, this would have been noteworthy.

We have clarified (lines 510-512):

However, in stable stratification and when the wind was directed from the open sea, the total amount of non-idealized profiles (1,335) was somewhat greater than the amount of idealized profiles (1,079), which is remarkable keeping in mind that non-idealized profiles on average appeared approximately 20% of the time in total, see Fig. 4(c).

**Ln 445,446,448: significant difference. If you make the statement it is significantly different then one should also add the results from an appropriate statistical test.**

In the box plots the notches mark the 95% confidence interval of the median. Notches are calculated according to McGill et al. (1978): notches = median $\pm$ 1.57 * (p75-p25)/sqrt(n) where p25 and p75 stands for the 25 and 75 percentiles respectively and n is the sample size. The significance level is based on assumption of a normal distribution of the data but is said to be rather robust also for other distributions. Further, the number of samples should be similar in the comparison of the medians, which cannot fully be said to the case in our analysis as the sample size for idealized profiles is much larger than for the non-idealized profiles (Table 1), which is of course important to further stress in our manuscript.

We have clarified and changed to the following (lines 525-531):

Starting by comparing the boxplots for transition profiles, weak LLJs and strong LLJs to the idealized profiles for the open sea sector in Fig. 9; as the notches do not overlap within any stability class there was a significant difference (5% significance level) in median spectral values, showing lower values for transition profiles and LLJs than for idealized profiles. The significance level comparing notches for the medians is based on an assumption of a normal distribution of the data for each box and that the number of samples are comparable (McGill et al. 1978). Thus, the significance of some of the results should be interpreted with great care as the sample size for idealized profiles in many cases outnumber the non-idealized profiles, see Table 1 for details.

For the Gotland sector, and similar to the results for the open sea sector, there was a significant difference in the normalized spectral values at the selected low frequency between transition profiles and LLJs compared

to idealized profiles in weakly unstable, near neutral and weakly stable conditions with the non-idealized profiles having lower spectral values. There was also a difference between the strong LLJs and the idealized profiles in stable stratification, with strong LLJs having significantly lower spectral values with 5% significance level. Although less data in the Östergarnsholm sector, there were indications of lower spectral values at the selected low frequency for wind profiles with an LLJ compared to idealized profiles.

**Ln 445,446,448: you mention that there were substantial differences between the spectra taken in the LLJ profiles and the normal profiles. But you do not describe how they differ (higher, lower, in which spectral ranges etc). Please expand.**

We have clarified this. See reply to the previous comment for the updated text (lines 525-531).

**Ln 466: in this respect I think it is better to refer to Kalverla et al (2018, https://onlinelibrary.wiley.com/doi/10.1002/we.2267 ) here since they focus on the evaluation of NWP models over sea (North Sea).**

Thank you for this suggestion. The reference have been changed to Kalverla et al. (2019).

Based on comments from Referee #2 we have also made some additional minor changes in the manuscript. An overall oversight for grammar and typos have also been conducted and we are ready to implement any additional adjustments required at this or later stages in the review process as is found necessary.

Sincerely,
Christoffer Hallgren and co-authors

**References**

Aird, J. A. et al. (2021). WRF-simulated low-level jets over Iowa: characterization and sensitivity studies. *Wind Energy Science* 6.4, pp. 1015–1030. DOI: 10.5194/wes-6-1015-2021.

Chapman, H. (2017). Performance test of the Pasquill stability classification scheme. The University of Wisconsin-Milwaukee (USA). MSc Thesis.

Finnigan, J. and Einaudi, F. (1981). The interaction between an internal gravity wave and the planetary boundary layer. Part II: Effect of the wave on the turbulence structure. *Quarterly Journal of the Royal Meteorological Society* 107.454, pp. 807–832. DOI: 10.1002/qj.49710745405.

Foken, T. (2006). 50 years of the Monin–Obukhov similarity theory. *Boundary-Layer Meteorology* 119.3, pp. 431–447. DOI: 10.1007/s10546-006-9048-6.

Gutiérrez-Loza, L. et al. (2019). Measurement of air-sea methane fluxes in the Baltic Sea using the eddy covariance method. *Frontiers in Earth Science* 7, p. 93. DOI: 10.3389/feart.2019.00093.

Högström, U. (1996). Review of some basic characteristics of the atmospheric surface layer. *Boundary-Layer Meteorology* 78.3, pp. 215–246. DOI: 10.1007/BF00120937.

Högström, U. et al. (2008). Momentum fluxes and wind gradients in the marine boundary layer — a multi-platform study. *Boreal Environment Research* 13.6, pp. 475–502. ISSN: 1797–2469. URL: https://helda.helsinki.fi/bitstream/handle/10138/235224/ber13-6-475.pdf.

Kaimal, J. C. et al. (1972). Spectral characteristics of surface-layer turbulence. *Quarterly Journal of the Royal Meteorological Society* 98.417, pp. 563–589. DOI: 10.1002/qj.49709841707.

Kalverla, P. C. et al. (2019). Evaluation of three mainstream numerical weather prediction models with observations from meteorological mast IJmuiden at the North Sea. *Wind Energy* 22.1, pp. 34–48. DOI: doi.org/10.1002/we.2267.

Kelly, M. and Gryning, S.-E. (2010). Long-term mean wind profiles based on similarity theory. *Boundary-layer meteorology* 136.3, pp. 377–390. DOI: 10.1007/s10546-010-9509-9.

Lehmann, A., Myrberg, K., and Höflich, K. (2012). A statistical approach to coastal upwelling in the Baltic Sea based on the analysis of satellite data for 1990–2009. *Oceanologia* 54.3, pp. 369–393. DOI: 10.5697/oc.54-3.369.

McGill, R., Tukey, J. W., and Larsen, W. A. (1978). Variations of Box Plots. *The American Statistician* 32.1, pp. 12–16. DOI: 10.1080/00031305.1978.10479236.

Nilsson, E. et al. (2012). Convective boundary-layer structure in the presence of wind-following swell. *Quarterly Journal of the Royal Meteorological Society* 138.667, pp. 1476–1489. DOI: 10.1002/qj.1898.

Obukhov, A. M. (1946). Turbulentnost'v temperaturnoj-neodnorodnoj atmosfere [Turbulence in an Atmosphere with a Non-uniform Temperature]. *Trudy Inst. Theor. Geofiz. AN SSSR* 1, pp. 95–115.

Pasquill, F. (1961). The estimation of the dispersion of windborne material. *Meteorological Magazine* 90, p. 33.

Prabha, T. V. et al. (2008). Influence of nocturnal low-level jets on eddy-covariance fluxes over a tall forest canopy. *Boundary-layer meteorology* 126.2, pp. 219–236. DOI: 10.1007/s10546-007-9232-3.

Román Cascón, C. (2010). Variability of turbulent fluxes (momentum, heat and CO2) during Upwelling conditions. A case study. Meteorology and Atmospheric Sciences, Uppsala University. MSc Thesis.

Rutgersson, A. et al. (2020). Using land-based stations for air–sea interaction studies. *Tellus A: Dynamic Meteorology and Oceanography* 72.1, pp. 1–23. DOI: 10.1080/16000870.2019.1697601.

Sahlée, E. et al. (2008). Spectra of CO 2 and water vapour in the marine atmospheric surface layer. *Boundary-layer meteorology* 126.2, pp. 279–295. DOI: 10.1007/s10546-007-9230-5.

Sanz Rodrigo, J. et al. (2015). Atmospheric stability assessment for the characterization of offshore wind conditions. *Journal of Physics: Conference Series*. Vol. 625. 1. IOP Publishing, p. 012044. DOI: 10.1088/1742-6596/625/1/012044.

Smedman, A.-S., Högström, U., and Hunt, J. (2004). Effects of shear sheltering in a stable atmospheric boundary layer with strong shear. *Quarterly Journal of the Royal Meteorological Society* 130.596, pp. 31–50. DOI: 10.1256/qj.03.68.

Smedman, A.-S. et al. (2009). Observational Study of Marine Atmospheric Boundary Layer Characteristics during Swell. *Journal of the atmospheric sciences* 66.9, pp. 2747–2763. DOI: 10.1175/2009JAS2952.1.

SMHI (2022). *Havsis [Sea ice]*. last access: 2022-01-18. URL: https://www.smhi.se/data/oceanografi/havsis.

Sorbjan, Z. and Grachev, A. A. (2010). An evaluation of the flux–gradient relationship in the stable boundary layer. *Boundary-layer meteorology* 135.3, pp. 385–405. DOI: 10.1007/s10546-010-9482-3.

Sproson, D. and Sahlée, E. (2014). Modelling the impact of Baltic Sea upwelling on the atmospheric boundary layer. *Tellus A: Dynamic Meteorology and Oceanography* 66.1, p. 24041. DOI: 10.3402/tellusa.v66.24041.

Stull, R. B. (1988). *An introduction to boundary layer meteorology*. Vol. 13. Springer Science & Business Media, Dordrecht, Boston, London. DOI: 10.1007/978-94-009-3027-8.

Sullivan, P. P. et al. (2008). Large-eddy simulations and observations of atmospheric marine boundary layers above nonequilibrium surface waves. *Journal of the Atmospheric Sciences* 65.4, pp. 1225–1245. DOI: 10.1175/2007JAS2427.1.

Svensson, N. et al. (2019). Measurements and Modelling of Offshore Wind Profiles in a Semi-Enclosed Sea. *Atmosphere* 10.4, p. 194. DOI: 10.3390/atmos10040194.

Zhang, S. et al. (2022). Coastal upwelling in the Baltic Sea from 2002 to 2020 using remote sensing data. Manuscript submitted for publication.

---

## Author Comment (AC2)

Dear Referee #2,

Thank you for your review of our manuscript and helpful comments! We have reviewed the manuscript according to your comments and corrected the manuscript point by point.

**General Comments**

**This study presents multi-year datasets combined to address differences in the wind profile characteristics throughout the year as a function of sea state, stability, season, and turbulence (via spectral analysis of inertial subrange and the encompassing scales). The different approaches toward conditioning the data made for an interesting read. The results were well discussed; however, there are areas that need further explanation and details before formal acceptance can be made. I would vote acceptance with minor reviews. The specific comments below are mostly related to the science within the paper and areas where additional explanation is needed. There were a lot of grammar/typo issues, and the reviewer likely missed many throughout the paper. It is suggested that the authors use the examples below in the technical corrections section to improve other areas of the paper beginning at the results section and continuing to the end.**

**Specific Comments**

**1. The authors investigate a multi-year dataset comprised of a tower, buoy, and a continuous wave lidar. Given the distribution of wind profiles and the discussion of "normal" v. "non-normal" profiles, where the latter includes profile types resembling LLJs as well as negative gradients, then it is strongly recommended that the title is adjusted so that it would not seem that the authors are specifically focusing on wind profiles of negative gradients. Afterall, that's only part of the story**

In accordance with your Specific Comment 2 below, we have changed the title to *Classification and properties of non-idealized coastal wind profiles – an observational study*.

**2. Comment of usage of "normal" v. "non-normal": I would argue that using "normal" v. "non-normal" to describe wind profiles is related to the region of interest. Certain regions, for example the great plains, experience LLJs during a fairly large proportion of days throughout the year. The same can be said with the great barrier jet off the coast of California. I would instead recommend the authors change "normal" to "idealized" and "non-normal" to "non-idealized" to describe departures from the ideal planetary boundary layer (PBL) model and the anticipated wind profile structure therein.**

Throughout the manuscript we have changed "normal" to "idealized" and "non-normal" to "non-idealized".

**3. Your section 3 is entitled "Methods" and yet it goes over the sites and the measurements used in addition to methods. I would change the title to reflect this. Something like "Site, Measurements, and Methods"**

Thank you for this comment. We have changed the title of Sect. 3 to Site, Measurements, and Methods.

**4. Lines 161-164: I don't quite understand this sentence fully. Are you also saying the buoy can measure turbidity? No results related to ocean turbidity is reported in your study. You do mention that wave properties have been evaluated against the turbulent state of the lower-atmosphere, but that is by making comparisons with the mast data. The sentence is confusing in other ways, too. Please improve your point here.** We have clarified this sentence, which now reads (line 158-159):

Only occasions when mast, lidar and wave buoy observations were simultaneously available were used in the analysis.

**5. Line 218: It's important to be sure that the linear interpolation is done over small time gaps. You probably should make that clear in the paper. Plus, I doubt that linear interpolation was done over large data gaps. You probably want to comment on any impact this could have on the results and what your gap threshold is. There are ways to approach this statistically as well if gaps are numerous and you have periods of similar turbulence intensity – with and without data gaps – then you can compare those data and apply statistical randomization such as what is used when solving for the p-values.**

We agree that this is an important point. Lines 181-191 now reads:

By selection criterias from Vitale et al. (2020) we assure to include in our analysis only half-hours when the longest duration of gaps are less than 3 minutes and further half-hours are selected to always contain more than 85% data coverage following the SevEr thresholds suggested by Vitale et al. (2020). In practice due to additional criteria the half-hour with lowest data availability included in our analysis was 92.8% (corresponding to a total of 130 s of 20 Hz data missing in the 30 min averaging period). However, only for 0.02% of the time (ten half-hours in total), the data availability was lower than 99%. Additionally, we used the homogeneity test of fluctuations and differenced data based on Chebyshev's inequality theorem in combination with the SevEr thresholds as suggested by Vitale et al. (2020) to avoid cases of large aberrant structural changes (e.g. sudden shifts in the mean value or changes in variance) which could lead to violation of the assumption of stationarity (Vitale et al. 2020). The longest gap duration within the high-frequency time series was typically short and only 1 half-hour had more than 12 consecutive data points missing in any of our selected 20 Hz time series for wind components and sonic temperature.

**6. Lines 345-347: It's hard to tell that the reason the wave age is higher is because of lower wind speeds. Clearly that could be born out the formulation that is used, but it is not shown clearly enough in Figure 3. The wind distribution over open seas is smaller during DJFM and the winds appear comparable when visually comparing the open sea sector distribution from DJFM with AMJJ. If you go ahead and multiply your wave age by U, then you should be able to compare the phase speeds between these periods to support this claim. Note your Figure 5 showing wind speed distributions and sea state. The AMJJ shows a much higher percentage of swell and more negative profiles, which weighs down the distribution toward lower wind speeds. The distribution, of course, can cannot easily be used to infer the wind rose distribution for open sector in figure 3.**

Thank you for this comment. In Fig. AC2.1 the normalised distributions of phase speed of the dominant waves in the three seasons DJFM, AMJJ and ASON are plotted. Fig. 3 in the manuscript indicates that the wave age (defined as $c_p/U$) is generally higher in AMJJ than in the other seasons and this is despite the fact that the phase speed is generally lower in this season (Fig. AC2.1). Thus we conclude that the higher wave age in AMJJ is due to the lower wind speeds. We have clarified this statement in the manuscript (lines 370-373):

In Fig. 3 the median wave age and the 25 and 75 percentiles of the wave age are presented for the open sea sector, showing slightly higher wave age on average during AMJJ. This is attributed to the typically lower wind speeds compared to the fall and winter seasons and is despite the fact that the phase speed of the dominant waves is in general lower in AMJJ than in other seasons (see supplement, Fig. S2).

[Figure]

Figure AC2.1: Normalized distributions of phase speed of the dominant waves in the three seasons DJFM, AMJJ and ASON when the wind is from the open sea sector. Bin size: $0.5 \text{ m s}^{-1}$.

**7. Lines 406-409: The mechanisms behind the LLJ were not determined in this paper which makes determining why LLJ-type profiles existed during different stability regimes and for different sectors difficult. The Gotland sector has more LLJs for neutral-to-unstable conditions while the open sea has more LLJs during stable conditions. The site that this data was collected is small, so it is likely to impact the vertical structure of the LLJ over the island but not cause it. Another tricky part of this analysis is differentiating between what the local stability is over land compared to the stability over the ocean. The air being advected over the land site would also impact stability, too, in addition to surface heating/cooling, and this relative impact on stability is not easily characterized with the Monin-Obukhov framework.**

We agree with all these points. Although different mechanisms behind LLJ formation are discussed in the manuscript it is difficult to determine the cause for every single case as many of the mechanisms might interact, especially at this coastal site where local effects from surrounding land areas have an impact on vertical structure of the wind field. In the new version of the Discussion we have made an attempt to better explain the different causes for LLJ formation and the impact that the coastal zone has on the stability measured at the site.

**8. Spectral Analysis: I know that the focus is mostly the inertial subrange and and frequency from which we depart from the inertial subrange into larger eddies. I agree with much of the discussion related to the spectral analysis, but I see that almost all "non-normal" profiles have lower energy containing eddies at the low frequency end. My guess is that one cause stems from dealing with much smaller sample sizes with the "non-normal" cases. It is interesting to note that only the negative profiles have examples where the largest energy containing eddies are larger than the "normal" profiles despite having the smallest wind shear and lightest wind conditions. Both Figure 8 and 9 show this. The production of turbulence is typically rooted in buoyancy or wind shear. Wind shear is small for the negative shear profiles so the thought is that buoyancy production should be responsible. However, the spectral energy is larger for stable conditions as well (Figure 9). Island meteorology and boundary layer internal advection is difficult. Can this be discussed or addressed in the paper?**

This is an important note and a complicated matter, which – as you also note – is partly outside the scope of our manuscript. We have added the following to the manuscript (lines 585-590):

As seen for the stable case in Fig. 8, there was a clear spectral gap separating the turbulent part of the spectra from larger scale non-turbulent motions such as atmospheric waves. While it could be argued that these low frequencies should be filtered out from the analysis (see e.g. Finnigan and Einaudi 1981), we decided to perform the analysis in a similar manner as e.g. Smedman et al. (2004), keeping all the turbulence data and analyse changes in the observed turbulence spectra, no matter if atmospheric waves were present or not. It can also be noted in Fig. 8 that the increase in spectral energy at the lowest frequencies was present not only for the case of an idealized wind profile, but also under non-idealized wind profile conditions.

**9. Line 538-539: There are studies that look into turbine wake characteristics as a function of stability. Look at these such examples: Zhan et. al. (2020) and Iungo et. al. (2014)**

Thank you for this comment. We have added the suggested references to the paragraph discussing wake characteristics under different atmospheric stabilities which now reads (lines 692-696):

Wind turbine wake behavior under different stabilities and for different wind speeds has been studied for both onshore (e.g. Iungo and Porté-Agel 2014; Zhan et al. 2020) and offshore wind farms (e.g. Platis et al. 2022) but more research is needed to accurately describe the wind resource within wind parks and how power production can be optimized using wake steering. The extent of the wake behind a wind turbine is likely to be very sensitive to LLJ conditions (e.g. Vollmer et al. 2017) as the potential to mix down momentum from above the jet core is greatly diminished.

**Technical Corrections (T and G will denote typos (T) v. grammar (G); grammatical suggestions will be marked by S)**

We thank you for taking the time to comment on all these technical corrections. All of the suggested changes have been implemented throughout the manuscript and based on comments from Referee #1 also other changes in the manuscript have been implemented, An overall oversight for grammar and typos have been conducted and we are ready to implement any additional adjustments required at this or later stages in the review process as is found necessary.

**T1 (Line 8). Change "...that the the zone with strong shear during low-level..." to "...that the strong shear zone of low-level..."**

**S1 (Lines 15-16). Change "...This variation, the wind shear, plays a..." to "The vertical structure of the wind profile (i.e., wind shear and wind veer) plays an...." With this statement you could lump the two sentences mentioning wind shear and veer together**

**G1 (Line 22). Change "...grow with an..." to "...grow at an..."**

**G2 (Line 23). Change "...scenario and that..." to "...scenario since it is anticipated that...."**

**S2 (Line 27). Change "...Either replace the commas at the beginning and end of "a high latitude semi-enclosed" with hyphens or insert "which is" after "The Baltic Sea,"**

**G3 (Line 30). Insert "of" between "GW" and "offshore"**

**S3 (Lines 32-33). Recommend replacing ".. and expansion has to be performed with care." To "...and therefore expansion must be handled with care."**

**T2 (Line 35). Change to "increases". Forgot to add "s"**

**G4 (Lines 36-37). Change "...are prone to have wind profiles with partly negative gradients that can occur under certain meteorological and..." to "...are prone to having wind profiles with partly negative gradients under certain meteorological and..."**

**S4 (Line 40). Suggest changing "Note that also wind..." to "Note, also, that wind..."**

**G5 (Line 44).** You use "both" to mention three additional factors. Both would apply to two additional factors. Perhaps change the following "...effects, both to assess the longevity of the turbines, the extension of the wake behind a single turbine and behind the park, and..." to effects; as well as to both to assess the longevity of the turbines, the extension of the wake behind a single turbine and behind the park, and..."

**S5 (Line 53).** Change "...what are the driving mechanisms for this." To "...what are the driving mechanisms that lead to turbulence production."

**G6 (Lines 58-59).** Change "In addition to this, not only the turbulent characteristics of the LLJs compared to normal profiles are analyzed...." to "In addition, not only are the turbulent characteristics of the LLJs compared to normal profiles..."

**S6 (Lines 69-70).** The sentence starting with "Already in 1957,..." is awkward. It just so happens that this is the year where this phenomenon was rigorously documented. I would consider starting this sentence with "One of the first proposed mechanisms related to the formation of..." You can fill in the rest.

**S7 (Line 71).** Instead of "During the evening and night,..." I would argue to use "During the evening transition,.."

**S8 (Lines 74-75).** The sentence beginning with "As a consequence..." is a bit of a mouthful. Perhaps make the following change: "...gradient force unbalanced, with a subsequent...." to "gradient force unbalanced. This imbalance subsequently leads to an acceleration of the wind: a process known as frictional decoupling."

**G7 (Line 80).** Change "As an effect..." to "As a result..." and change "compared to a water..." to "...compared to the water..."

**S9 (Line 86).** Recommend changing "at least if the swell and the wind direction are aligned which is the most studied case." to "...if the wind is approximately aligned with the swell direction"

**T3/G8 (Lines 91-92).** Change "...on theses matters, but for an introduction to uncommon wind profiles over the Baltic Sea and North Sea, we refer to Kettle (2014) and Moller et. al. (2020)." To "...these different wind profile types. We refer to studies by Kettle (2014) and Moller et. al. (2020) for a description of less common wind profiles of this type."

**G9 (Line 119).** Change "In a following..." to "In the following...."

**T4 (Line 169).** Remove "is" after "as" and before "possible"

**G10 (Line 185).** Do not need comma after "properly" and you remove "those" after "remove"

**G11 (Line 210).** I was confused with the wording here: "...site of especially swell conditions." Were you trying to say "...site, especially during swell conditions."

**G12 (Line 228).** Change "on" after "laser" and before "a" to "at"

**G13 (Line 246).** I would change the sentence starting with "In this additional..." to something like "Application of additional quality controls led to the removal of 6.7% of the data."

**G14 (Line 253).** Replace "on" between "depth" and "buoy" with "at"

**General comment:** When introducing a variable, say, $C_p$ , it is common to stick two commas between the variable.

**G15 (Line 257).** Replace "in" between "10.4 m" and "the" with "from"

**G16 (Line 258).** Don't need a comma between "sector" and "since"

**G17 (263). Replace "on" between "mast" and "Ostergarnsholm" with "at". I've seen this for multiple occasions. I would make sure that this checked elsewhere.**

Sincerely,
Christoffer Hallgren and co-authors

**References**

Finnigan, J. and Einaudi, F. (1981). The interaction between an internal gravity wave and the planetary boundary layer. Part II: Effect of the wave on the turbulence structure. *Quarterly Journal of the Royal Meteorological Society* 107.454, pp. 807–832. DOI: 10.1002/qj.49710745405.

Iungo, G. V. and Porté-Agel, F. (2014). Volumetric lidar scanning of wind turbine wakes under convective and neutral atmospheric stability regimes. *Journal of Atmospheric and Oceanic Technology* 31.10, pp. 2035–2048. DOI: 10.1175/JTECH-D-13-00252.1.

Platis, A. et al. (2022). The Role of Atmospheric Stability and Turbulence in Offshore Wind-Farm Wakes in the German Bight. *Boundary-Layer Meteorology* 182.3, pp. 441–469. DOI: 10.1007/s10546-021-00668-4.

Smedman, A.-S., Högström, U., and Hunt, J. (2004). Effects of shear sheltering in a stable atmospheric boundary layer with strong shear. *Quarterly Journal of the Royal Meteorological Society* 130.596, pp. 31–50. DOI: 10.1256/qj.03.68.

Vitale, D. et al. (2020). A robust data cleaning procedure for eddy covariance flux measurements. *Biogeosciences* 17, pp. 1367–1391. DOI: 10.5194/bg-17-1367-2020.

Vollmer, L. et al. (2017). A wind turbine wake in changing atmospheric conditions: LES and lidar measurements. *Journal of Physics: Conference Series*. Vol. 854. 1. IOP Publishing, p. 012050. DOI: 10.1088/1742-6596/854/1/012050.

Zhan, L., Letizia, S., and Valerio Iungo, G. (2020). LiDAR measurements for an onshore wind farm: Wake variability for different incoming wind speeds and atmospheric stability regimes. *Wind Energy* 23.3, pp. 501–527. DOI: 10.1002/we.2430.